# Generalized Fast Exact Conformalization

**Diyang Li**
Cornell University
diyang01@cs.cornell.edu

## Abstract

Conformal prediction converts nearly any point estimator into a prediction interval under standard assumptions while ensuring valid coverage. However, the extensive computational demands of full conformal prediction are daunting in practice, as it necessitates a comprehensive number of trainings across the entire latent label space. Unfortunately, existing efforts to expedite conformalization often carry strong assumptions and are developed specifically for certain models, or they only offer approximate solution sets. To address this gap, we develop a method for fast exact conformalization of generalized statistical estimation. Our analysis reveals that the structure of the solution path is inherently piecewise smooth, and indicates that utilizing second-order information of difference equations suffices to approximate the entire solution spectrum arbitrarily. We provide a unified view that not only encompasses existing work but also attempts to offer geometric insights. Practically, our framework integrates seamlessly with well-studied numerical solvers. The significant speedups of our algorithm as compared to the existing standard methods are demonstrated across numerous benchmarks.

## 1 Introduction

In modern algorithmic practice, quantifying uncertainty is crucial for accurate and reliable model predictions. Conformal prediction [1] serves as a powerful statistical tool that leverages the observed data to construct prediction intervals containing the outcome with a predefined probability level. It enjoys model-free coverage guarantee regardless of the underlying distribution of the data. In recent years, conformal prediction has gained increasing attention from the community of machine learning [2–5], data mining [6–8] and computer vision [9, 10]. This growing interest is attributed to the attractive properties that it operates under the assumption of exchangeability, which is a weaker condition than independence and identical distribution, allowing for a wider range of applications in real-world scenarios where data may not meet strict statistical assumptions. Meanwhile, conformal prediction can be combined with almost any existing point estimators, even when the model is potentially misspecified [5].

While exhibits appealing properties, the application of conformal prediction often comes at a high computational cost [11, 12]. Kindly note that in this paper we refer to the full conformal prediction that does *not* discard training points as opposed to the split conformal prediction, as the latter involves only one single fitting. From a numerical perspective, when constructing a conformal prediction set, one needs to exhaustively search all points (potential candidates) in the label space, where for each point the learning model needs to be refitted and the conformity score needs to be re-calculated. In many scenarios like regression, the number of possible candidates is infinite as the latent label can take an uncountable number of possible values. Conventional conformal prediction works by a grid-search type method to loop over the label space [13, 14], which discretize the interval of interest and subsequently solve a sequence of individual optimization subproblems. To improve such brute-force approach, there have been many efforts in community that devoted to develop better algorithms for computing the prediction set. A natural idea is to generate the set of all solutions

38th Conference on Neural Information Processing Systems (NeurIPS 2024).

Table 1: Representative related work, which are instances of *generalized parametric estimations*.

| Model | Reference | Exact | Loss | Regularizer | Constrained | Path Structure |
|---|---|---|---|---|---|---|
| Least Squares | [15] | ✓ | Quadratic | \ | ✗ | Piecewise Linear |
| Ridge Regression | [16] | ✓ | Quadratic | $\|\mathbf{w}\|^2$ | ✗ | Piecewise Linear |
| Empirical Risk Minimization | [17] | ✗ | Convex | Convex | ✗ | Piecewise Smooth |
| Elastic Net | [12] | ✓ | Quadratic | $\|\mathbf{w}\|_1$ | ✗ | Piecewise Linear |
| Generalized Lasso Regression | [18] | ✗ | Convex | $\|\mathbf{w}\|_1$ | ✗ | Piecewise Linear |
| General Formulation (**ours**) | Section 4 | ✓ | $PC^r$ | $PC^r$ | ✓ | Piecewise Smooth |

indexed by the latent label candidate using numerical continuation (*a.k.a.* homotopy) method, and we name this set of optimal solutions as (exact) *solution path*, as illustrated in Figure 1.

Despite extensive theoretical and empirical efforts, the understanding of conformalization path remains rather deficient. For some simple cases, closed-form characterizations of conformal prediction sets are available, such as $k$-nearest neighbors, least squares regression [15], and ridge regression [16] with quadratic loss. The study by [12] presents an exact solution path for conformalized Lasso and elastic net using their statistical property and $\ell_1$-sparsity analysis. Investigations into more general objectives as discussed in [17, 18] incorporate linear interpolation to approximate the intrinsic piecewise smooth structure, yielding prediction sets lacking of finite-sample calibration. In other terms, [17, 18] offer only an upper bound for the approximation error and fail to control the degree of approximation relative to the optimal solutions. We summarize these relevant prior studies in Table 1. Given that existing exact algorithms are either tailored to a select few models, or turn out to be grid-search type approaches that take a very black-box approach, it prompts the following question:

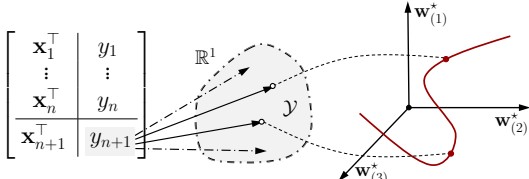

Figure 1: Diagram of setting on fast exact conformalization, where label candidate $y_{n+1}$ should loop over the whole latent label space $\mathcal{Y}$ and each possible $y_{n+1} \in \mathbb{R}^1$ corresponds to one point on the (red) solution path. Conventional practice is to refit the model on each new $y_{n+1}$ while we employ the path-following algorithm to obtain the whole solution spectrum within 1 execution.

> *Can we "open the black-box" by developing a methodology*
> *that better exploits the structure of the path?*

We answer this question positively by introducing a differential equation perspective to analyze the ground truth solution path, which enables us to better reveal and exploit the fundamental path structure. This more profound understanding enables us to build more generalized conformalization algorithm and present improved computational guarantees.

## 1.1 Our contributions

The main contributions brought by this paper are summarized as follows.

**Generalizable framework** This study aims to extend the application of fast exact conformalization into generalized statistical estimation. We relax the assumptions by considering a cost function that is no longer globally differentiable, but rather *piecewise differentiable*, while introducing constraints on the weight vector to fit more statistical models. In our analysis, the Clarke subdifferential of the objective is derived, and the set of nonsmooth points of path can essentially be described as a level set of certain smooth functions. By adopting a reparameterization regime, our framework effectively broadens the scope of several existing algorithms in Table 1 and provides a unified view of them.

**Theoretical insights** We analyze the underlying structure of path through the first-order optimality conditions of the regularized problem, identifying sufficient conditions for a local path to be smooth around a given point and reveal that the structure of the path is inherently piecewise smooth. More precisely, if conditions are met, then the path is locally the projection of a higher-dimensional smooth manifold onto the optimization space, thus offering preliminary geometric intuitions. Our

investigation further suggests that leveraging the second-order information of difference equations can approximate the solution spectrum arbitrarily.

**Practical efficiency** Practically, our framework is not only straightforward to implement but also computationally efficient. With theoretical analysis, we present explicit expressions for the gradient flows of objective, which is homogenized and well aligned with standard forms used by mainstream numerical libraries for ordinary differential equations (ODE), thereby easing the programming efforts. When crossing potential kinks (or nonsmooth points), the computations are facilitated by boundary conditions pre-set in the numerical solver. Notably, our algorithm eliminates the need for extensive iterations for computing the entire solution spectrum, contrasting sharply with conventional baselines. In experiments, we demonstrate the significant computational speed advantages of our algorithm over existing baselines, without compromising on accuracy.

## 2 Background

**Notation** For a set $\mathcal{J} \subseteq \mathbb{R}^p$, we denote by $\mathrm{cl}(\mathcal{J})$ the closure and by $\mathrm{int}(\mathcal{J})$ the interior of $\mathcal{J}$ w.r.t. the natural topology on $\mathbb{R}^p$. Define $\mathcal{J}(i)$ the $i$-th element of $\mathcal{J}$ and $\mathrm{conv}(\mathcal{J})$ be the convex hull of $\mathcal{J}$, or $\mathrm{conv}(\mathcal{J}) = \{v : v = \sum_{i=1}^k \hat{\theta}_i u_i, u_i \in \mathcal{J}, \hat{\theta}_i \in \mathbb{R}^{>0}, \sum_{i=1}^k \hat{\theta}_i = 1\}$. The $\mathbf{w}_{(i)}$ is the $i$-th element of vector $\mathbf{w}$ and $\mathbb{I}(\cdot)$ is an indicator function. Let $\dot{z}(t)$ be the derivative $\frac{dz(t)}{dt}$ of function $z(t)$. Let $\mathbf{O}$ be the zero matrix and $\mathbb{P}\{\cdot\}$ be the event probability. The $\mathrm{sgn}(\cdot)$ is the sign function $\mathrm{sgn}(x) = \frac{x}{|x|}(x \neq 0)$ or $0$ $(x = 0)$ that applied entrywise.

### 2.1 Model and assumptions

Given the dataset $\{\mathbf{x}_i, y_i\}_{i=1}^n$, where $\mathbf{x}_i \in \mathbb{R}^p$ is the sample (covariate) and $y_i$ is the $i$-th label (response) live in label space $\mathcal{Y} \subseteq \mathbb{R}^1$. We consider the generalized parametric estimation problem

$$
\begin{aligned}
\mathbf{w}^\star \in \arg\min_{\mathbf{w}} \quad & \sum_{i=1}^n L_i \left( y_i, \eta_{\mathbf{w}}(\mathbf{x}_i) \right) + \sum_{j=1}^m \lambda_j \Omega_j(\mathbf{w}) \\
s.t. \quad & g_i(\mathbf{w}) = 0, \quad 1 \leq i \leq r, \\
& h_j(\mathbf{w}) \leq 0, \quad 1 \leq j \leq s,
\end{aligned}
\tag{1}
$$

where $\eta_{\mathbf{w}}$ is the model prediction function, $L_i$ is the loss and $\Omega_j$ is the regularizer. The $g_i$, $h_j$ are constraints on $\mathbf{w}$, and parameter $\lambda_j \in \mathbb{R}^{>0}$ controls the degree of regularity. In the following, we will define the piecewise differentiability and state main assumptions that used throughout this work.

**Definition 1** (*$PC^r$ Function*). *Let $f : U \to \mathbb{R}$ be continuous on the open set $U \subseteq \mathbb{R}^p$ and $f_i : U \to \mathbb{R}$, $i \in \{1, \ldots, k\}$ be a set of $r$-times continuously differentiable (or $C^r$) functions for $r \in \mathbb{N} \cup \{\infty\}$. If $f(x) \in \{f_i(x)\}_{i \in \{1,\ldots,k\}}$ holds for all $x \in U$, then $f$ is an $r$-times piecewise continuously differentiable (or $PC^r$) function. The $\{f_1, \ldots, f_k\}$ is a set of selection functions of $f$.*

When working with $PC^r$-functions in a local sense, it is useful to only consider the selection functions that have an impact on the local behavior around a given point.

**Definition 2** (*Essentially Active Set*). *Let $f : U \to \mathbb{R}$ be a $PC^r$ function on the open set $U \subseteq \mathbb{R}^p$ with a set of selection functions $\{f_1, \ldots, f_k\}$. Denote $I_K = \{1, \ldots, k\}$. Then given a $x_1 \in U$, $I_f^a(x_1) \triangleq \{i \in I_K : f(x_1) = f_i(x_1)\}$ is called the active set at $x_1$, and the $I_f^e(x_1) \triangleq \{i \in I_K : x_1 \in \mathrm{cl}(\mathrm{int}(\{x_2 \in U : f(x_2) = f_i(x_2)\}))\}$ is the essentially active set at $x_1$.*

**Assumption 1.** *We assume that $L_i$ and $\Omega_j$ in (1) are $PC^r$ functions each with a set of selection functions $\bigcup_{k \in I_{L_i}^e} \{D_{L_i}^k\}$ and $\bigcup_{k \in I_{\Omega_j}^e} \{D_{\Omega_j}^k\}$, respectively.*

**Assumption 2.** *We assume that $\Omega_j$ and $L_i$ are non-differentiable at $\mathbf{w}$ with multiple active selection functions, where $j \in \{1, \ldots, m\}$, $i \in \{1, \ldots, n + 1\}$. We further assume that $I_{\Omega_j}^a(\mathbf{w}) \equiv I_{\Omega_j}^e(\mathbf{w}), I_{L_i}^a(\mathbf{w}) \equiv I_{L_i}^e(\mathbf{w})$ holds for all $\mathbf{w}$ considered and all $\Omega_j, L_i$ in the following.*

Assumption 2 ensures that all selection functions we consider are actually relevant for the representation of $\Omega_j, L_i$. *i.e.*, it does not matter if we consider the active or the essentially active set in the underlying optimizing space, which allows for an easier representation of $D_{\Omega_j}^k$ and $D_{L_i}^k$.

## 2.2 Conformal prediction

**Definition 3** (Symmetrical Algorithm). *A deterministic algorithm $A : (x_1, \ldots, x_n) \to A^\star$ is symmetric if for any permutation $\tau$ of $\{1, \ldots, n\}$, $A(x_1, \ldots, x_n) \overset{a.s.}{=} A(x_{\tau(1)}, \ldots, x_{\tau(n)})$.*

**Definition 4** (Conformity Score). *The conformity score function $\mathcal{A}$, symmetric in its first $n$ inputs, is defined as $\mathcal{A}(\vec{\mathbf{x}}_1, \ldots, \vec{\mathbf{x}}_n; \vec{\mathbf{x}}_{n+1}): \mathbb{R}^{(p+1) \times (n+1)} \to \mathbb{R}$, where $\vec{\mathbf{x}}_i \triangleq (\mathbf{x}_i, y_i)$.*

Conformal prediction starts from a conventional model fitting stage, followed by the evaluation of conformity score and the construction of prediction set. The score function $\mathcal{A}$ serves as a measure of deviation or conformity, assessing the extent to which the new input $\mathbf{x}_{n+1}$ aligns with the previously fitted model. A higher conformity score indicates a better match between $\mathbf{x}_{n+1}$ and the model [1, 14]. For a new instance $\mathbf{x}_{n+1}$ where the prediction region is desired, the conformalization method operates by assigning a $p$-value to each latent $y_{n+1} \in \mathcal{Y}$, formalized as

$$\hat{p}_{y_{n+1}} = 1 - \frac{1}{n+1} \sum_{i=1}^{n+1} \mathbb{I}\left(\mathcal{A}_i \geq \mathcal{A}_{n+1}\right), \tag{2}$$

where $\mathcal{A}_i \triangleq \mathcal{A}(\vec{\mathbf{x}}_1, .., \vec{\mathbf{x}}_{i-1}, \vec{\mathbf{x}}_{i+1}, .., \vec{\mathbf{x}}_{n+1}; \vec{\mathbf{x}}_i)$. Specifically, in density estimation, $\mathcal{A}$ is defined as $\eta_{\mathbf{w}^\star}(\mathbf{x}_{n+1})$, where $\eta_{\mathbf{w}^\star}$ is the density function estimated from the augmented dataset. For regression tasks, $\mathcal{A}$ might be set as $-|y_{n+1} - \eta_{\mathbf{w}^\star}(\mathbf{x}_{n+1})|$, with $\eta_{\mathbf{w}^\star}$ being the regression function trained by the dataset including $n + 1$ samples. Under the assumption of exchangeability among the pairs $\{\mathbf{x}_i, y_i\}_{i=1}^{n+1}$, the $\hat{p}_{y_{n+1}}$ returned by (2) has been demonstrated to be statistically valid [13, 14]. To generate the prediction set $\Gamma(\cdot)$, one thresholds these $p$-values at a prescribed error level $\alpha \in (0, 1)$, resulting in

$$\Gamma(\mathbf{x}_{n+1}) = \left\{y_{n+1} : \hat{p}_{y_{n+1}} \geq \alpha\right\}. \tag{3}$$

**Theorem 1.** *[19] Suppose that $\{\mathbf{x}_i, y_i\}_{i=1}^{n+1}$ are exchangeable and the fitting algorithm $A$ is symmetric. Conformal prediction applied on $\{\mathbf{x}_i, y_i\}_{i=1}^{n} \cup \{\mathbf{x}_{n+1}\}$ outputs a set $\Gamma(\cdot)$ such that*

$$\mathbb{P}\left\{y_{n+1}^\star \in \Gamma(\mathbf{x}_{n+1})\right\} \geq 1 - \alpha, \tag{4}$$

*where $y_{n+1}^\star$ is the ground truth $(n + 1)$-th label.*

Theorem 1 (*a.k.a.* coverage guarantee) requires only exchangeability of input data and symmetry of the conformity score function, which are met by nearly all prevalent model fitting algorithms. In alignment with the analysis taken in prior research, we treat $y_{n+1} := y_{n+1}(z)$ as a function of scalar variable $z$. We utilize it to facilitate the traversal of $y_{n+1}$ across the entire label space $\mathcal{Y}$, and compute the homotopy solution path $\{\mathbf{w}^\star(z) : z_{\min} \leq z \leq z_{\max}\}$ (also shown in Figure 1). We assume that $y_{n+1}(\cdot): [z_{\min}, z_{\max}] \to \mathcal{Y}$ is continuously differentiable in terms of $z$ for simplicity.

# 3 Main results

We present our main results for fast exact conformalization. The discussions here focus on positive $z$,[1] but the derivation extends easily to include negative values as well, which will be discussed later.

## 3.1 Surrogate function

**Lemma 1.** *Let $f: U \to \mathbb{R}$ be a $PC^r$-function on the open set $U \subseteq \mathbb{R}^p$ and let $C^r$-functions $\{f_1, \ldots, f_k\}$ be a set of selection functions of $f$. Then for any $x \in U$, there exists an open neighborhood $U' \subseteq U$ of $x$ on which $f$ is also a continuous selection of $\{f_i : i \in I_f^e(x)\}$.*

Drawing insights from Lemma 1, we minimize the surrogate function of (1) as

$$\min_{\mathbf{w} \in \mathbb{R}^p} \mathcal{E}_z(\mathbf{w}) := \sum_{i=1}^{n+1} L_i\left(y_i, \eta_{\mathbf{w}}(\mathbf{x}_i)\right) + \sum_{j=1}^{m} \lambda_j \Omega_j(\mathbf{w}) + \rho \sum_{i=1}^{r} |g_i(\mathbf{w})| + \rho \sum_{j=1}^{s} \max\left\{0, h_j(\mathbf{w})\right\}, \tag{5}$$

---

[1]For technicality reasons, we enforce $z > 0$ even the limit $\mathbf{w}^\star(0^+) \triangleq \lim_{z \to 0^+} \mathbf{w}^\star(z)$ might be well-defined.

where $\rho \in \mathbb{R}^{>0}$ and $y_{n+1} = y_{n+1}(z)$. This definition of $\mathcal{E}_z(\mathbf{w})$ is meaningful regardless of whether the contributing functions are convex. Denote model estimation terms $\sum_i L_i + \sum_j \lambda_j \Omega_j$ as $\mathbb{L}_M(\mathbf{w}|z)$, and the minimizer of (5) as $\mathbf{w}^\star(z)$. It is interesting to compare $\mathcal{E}_z(\mathbf{w})$ to the Lagrangian function

$$\mathcal{L}_z(\mathbf{w}) := \mathbb{L}_M(\mathbf{w}|z) + \sum_{i=1}^{r} \tilde{\lambda}_i g_i(\mathbf{w}) + \sum_{j=1}^{s} \tilde{\mu}_j h_j(\mathbf{w}), \tag{6}$$

which captures the behavior of $\mathbb{L}_M(\cdot)$ near the optimum. At a constrained minimum $\mathbf{w}^\star$, the Lagrangian satisfies the stationarity condition $\nabla \mathcal{L}(\mathbf{w}^\star) = \mathbf{0}$; its inequality multipliers $\tilde{\mu}_j$ are nonnegative and satisfy the complementary slackness $\tilde{\mu}_j h_j(\mathbf{w}^\star) = 0$. In the penalized (5), one usually takes

$$\rho > \max\{|\tilde{\lambda}_1|, \ldots, |\tilde{\lambda}_r|, \tilde{\mu}_1, \ldots, \tilde{\mu}_s\}, \tag{7}$$

which creates the favorable circumstances: (i) $\mathcal{L}_z(\mathbf{w}) \leq \mathcal{E}_z(\mathbf{w})$ for all $\mathbf{w}$, (ii) $\mathcal{L}_z(\mathbf{w}) \leq \mathbb{L}_M(\mathbf{w}|z) = \mathcal{E}_z(\mathbf{w})$ for all feasible $\mathbf{w}$, (iii) $\mathcal{L}_z(\mathbf{w}^\star) = \mathbb{L}_M(\mathbf{w}^\star|z) = \mathcal{E}_z(\mathbf{w}^\star)$ with profound consequences.

**Theorem 2.** *Our surrogate function $\mathcal{E}_z(\mathbf{w})$ is increasing in $\rho$. Furthermore, $\mathcal{E}_z(\mathbf{w})$ is strictly convex for one $\rho > 0$ if and only if it is strictly convex for all $\rho > 0$. Likewise, it is coercive for one $\rho > 0$ if and only if it is coercive for all $\rho > 0$. Finally, if $\mathbb{L}_M(\cdot)$ is strictly convex (or coercive), then $\mathcal{E}_z(\mathbf{w})$ for all $\rho$ are strictly convex (or coercive).*

Given Theorem 2, several classical results [20] state that minimizing $\mathcal{E}_z(\mathbf{w})$ is effective in minimizing $\mathbb{L}_M(\cdot)$ subject to the constraints if we choose $\rho$ by (7).

## 3.2 Conformal path characterization

**Definition 5** (Clarke Subdifferential). *For continuous function $f : U \to \mathbb{R}$ on the open set $U \subseteq \mathbb{R}^p$, let $\Theta \subseteq U$ be the set of points in which $f$ is not differentiable. The Clarke subdifferential of $f$ at $x \in U$ is defined as*

$$\partial f(x) \triangleq \mathrm{conv}(\{\phi \in \mathbb{R}^p : \exists\{x_i\}_{i=1}^{\infty} \in \mathbb{R}^p \backslash \Theta \text{ with } \lim_{i \to \infty} x_i = x, \lim_{i \to \infty} \nabla f(x_i) = \phi\}).$$

Specifically, if $f$ is continuously differentiable in $x$, $\partial f(x) = \{\nabla f(x)\}$. While $PC^r$-functions are generally non-smooth, we can use the Clarke subdifferential to obtain first-order optimality of (5),

$$\sum_{i=1}^{n} \sum_{k \in I_{L_i}^a(\mathbf{w}^\star)} \hat{\theta}_{L_i}^k(\mathbf{w}^\star) \nabla D_{L_i}^k(y_i, \eta_{\mathbf{w}^\star}(\mathbf{x}_i)) + \sum_{k \in I_{L_{n+1}}^a(\mathbf{w}^\star)} \hat{\theta}_{L_{n+1}}^k(\mathbf{w}^\star) \nabla D_{L_{n+1}}^k(y_{n+1}(z), \eta_{\mathbf{w}^\star}(\mathbf{x}_{n+1})) \triangleq \mathbf{D}'(\mathbf{w}^\star),$$

$$\rho \sum_{i=1}^{r} \hat{\theta}_{g_i} \nabla g_i(\mathbf{w}^\star) + \rho \sum_{j=1}^{s} \hat{\theta}_{h_j} \nabla h_j(\mathbf{w}^\star) + \sum_{j=1}^{m} \sum_{k \in I_{\Omega_j}^a(\mathbf{w}^\star)} \lambda_j \hat{\theta}_{\Omega_j}^k(\mathbf{w}^\star) \nabla D_{\Omega_j}^k(\mathbf{w}^\star) + \mathbf{D}'(\mathbf{w}^\star) = \mathbf{0}, \tag{8}$$

where $\hat{\theta}_{\Omega_j}^k, \hat{\theta}_{L_i}^k$ is the $k$-th auxiliary parameter for convex hull of each $\Omega_j, L_i$. The (8) is accompanied by the active sets conditions and the subdifferentials conditions, rewritten in detail as

$$\begin{aligned} D_{L_i}^k(\mathbf{w}^\star) - D_{L_i}^{l_i}(\mathbf{w}^\star) &= 0, \ \forall k \in I_{L_i}^a(\mathbf{w}^\star) \backslash \{l_i\}, \ \forall i \in \bar{I}_{L_i}^e \\ D_{\Omega_j}^k(\mathbf{w}^\star) - D_{\Omega_j}^{r_j}(\mathbf{w}^\star) &= 0, \ \forall k \in I_{\Omega_j}^a(\mathbf{w}^\star) \backslash \{r_j\}, \ \forall j \in \bar{I}_{\Omega_j}^e \\ \sum_{k \in I_{L_i}^a(\mathbf{w}^\star)} \hat{\theta}_{L_i}^k(\mathbf{w}^\star) - 1 &= 0, \ \hat{\theta}_{L_i}^k(\mathbf{w}^\star) \geq 0, 1 \leq i \leq n+1 \\ \sum_{k \in I_{\Omega_j}^a(\mathbf{w}^\star)} \hat{\theta}_{\Omega_j}^k(\mathbf{w}^\star) - 1 &= 0, \ \hat{\theta}_{\Omega_j}^k(\mathbf{w}^\star) \geq 0, \ 1 \leq j \leq m \end{aligned} \tag{9}$$

where $r_j, l_i$ is randomly selected from $I_{\Omega_j}^a, I_{L_i}^a$ and being fixed, with coefficients satisfying $\hat{\theta}_{g_i} \in$
$\begin{cases} \{-1\} & g_i(\mathbf{w}) < 0 \\ [-1, 1] & g_i(\mathbf{w}) = 0 \\ \{1\} & g_i(\mathbf{w}) > 0 \end{cases}, \hat{\theta}_{h_j} \in \begin{cases} \{0\} & h_j(\mathbf{w}) < 0 \\ [0, 1] & h_j(\mathbf{w}) = 0 \\ \{1\} & h_j(\mathbf{w}) > 0 \end{cases}$. In this work, we specialize to the case where the

constraint functions $g_i$ $(1 \le i \le r)$ and $h_j$ $(1 \le j \le s)$ are *affine*, *i.e.*, the gradients $\nabla g_i(\mathbf{w}), \nabla h_j(\mathbf{w})$ are constant.[2] We define $g_i$ and $h_j$ as constraint residuals $g_i(\mathbf{w}) := \mathbf{v}_i^\top \mathbf{w} - d_i$, $h_j(\mathbf{w}) := \boldsymbol{\omega}_j^\top \mathbf{w} - e_j$.

We keep track of the following index sets determined by signs of constraint residuals:

$$\mathcal{N}_{\mathrm{E}} = \{i : g_i(\mathbf{w}) = \mathbf{v}_i^\top \mathbf{w} - d_i < 0\}, \mathcal{Z}_{\mathrm{E}} = \{i : g_i(\mathbf{w}) = \mathbf{v}_i^\top \mathbf{w} - d_i = 0\}, \mathcal{P}_{\mathrm{E}} = \{i : g_i(\mathbf{w}) = \mathbf{v}_i^\top \mathbf{w} - d_i > 0\},$$
$$\mathcal{N}_{\mathrm{I}} = \{j : h_j(\mathbf{w}) = \boldsymbol{\omega}_j^\top \mathbf{w} - e_j < 0\}, \mathcal{Z}_{\mathrm{I}} = \{j : h_j(\mathbf{w}) = \boldsymbol{\omega}_j^\top \mathbf{w} - e_j = 0\}, \mathcal{P}_{\mathrm{I}} = \{j : h_j(\mathbf{w}) = \boldsymbol{\omega}_j^\top \mathbf{w} - e_j > 0\}. \tag{10}$$

To characterize the path, we further introduce a *reparameterization* in terms of an auxiliary variable $t \ge 0$ (thought of as *time*), whereby for a given $\mathbb{T} > 0$ we introduce functions $z(\cdot) : [0, \mathbb{T}] \to [z_{\min}, z_{\max}]$ and $\xi(\cdot) : [z_{\min}, z_{\max}] \to \mathbb{R}$ such that: $\xi(\cdot)$ is Lipschitz, $z(\cdot)$ is differentiable on $(0, \mathbb{T})$, and we have $\dot{z} = \xi(z(t))$ for all $t \in (0, \mathbb{T})$. In a slight abuse of notation, we define the path w.r.t. $t$ as $\left\{ \mathbf{w}^\star(t) \triangleq \mathbf{w}^\star(z(t)) : t \in [0, \mathbb{T}] \right\}$. To enhance structural clarity, we denote $n_{\mathcal{Z}} = |\mathcal{Z}_{\mathrm{E}} \cup \mathcal{Z}_{\mathrm{I}}|$ and represent the certain matrix inversion as

$$\left[ \begin{array}{cc} \tilde{\mathbf{H}}(\mathbf{w}^\star | z) & \mathbf{U}_{\mathcal{Z}}^\top \\ \mathbf{U}_{\mathcal{Z}} & \mathbf{O}_{n_{\mathcal{Z}} \times n_{\mathcal{Z}}} \end{array} \right]^{-1} = \left[ \begin{array}{cc} \mathbf{P}(\mathbf{w}^\star | z) & \mathbf{Q}(\mathbf{w}^\star | z) \\ \mathbf{Q}^\top(\mathbf{w}^\star | z) & \mathbf{R} \end{array} \right], \tag{11}$$

where the rows of matrix $\mathbf{U}_{\mathcal{Z}}$ are the constant differentials, $\mathbf{v}_i^\top$ for $i \in \mathcal{Z}_{\mathrm{E}}$ and $\boldsymbol{\omega}_j^\top$ for $j \in \mathcal{Z}_{\mathrm{I}}$, and

$$\tilde{\mathbf{H}}(\mathbf{w}^\star | z) \triangleq \sum_{i=1}^{n} \sum_{k \in I_{L_i}^a(\mathbf{w}^\star)} \hat{\theta}_{L_i}^k(\mathbf{w}^\star) \nabla^2 D_{L_i}^k(y_i, \eta_{\mathbf{w}^\star}(\mathbf{x}_i)) + \sum_{j=1}^{m} \sum_{k \in I_{\Omega_j}^a(\mathbf{w}^\star)} \lambda_j \hat{\theta}_{\Omega_j}^k(\mathbf{w}^\star) \nabla^2 D_{\Omega_j}^k(\mathbf{w}^\star)$$
$$+ \sum_{k \in I_{L_{n+1}}^a(\mathbf{w}^\star)} \hat{\theta}_{L_{n+1}}^k(\mathbf{w}^\star) \nabla^2 D_{L_{n+1}}^k(y_{n+1}(z), \eta_{\mathbf{w}^\star}(\mathbf{x}_{n+1})). \tag{12}$$

Now we are fully equipped to unveil the structure of homotopy path, as shown in next subsection.

### 3.3 Piecewise smooth structure & Unified view

To describe the structure, we define the kink (non-smooth point) as the point that $\hat{\theta}_{\Omega_j}^k$, $\hat{\theta}_{L_i}^k$ hit the restriction bound in (9), or set in (10) is violated so the entire structure changes.

**Theorem 3.** *Given an optimum $(\mathbf{w}_0^\star, z_0)$ at $t_0$, and assume that $t_0$ is not a kink, then there exists an open neighborhood of $t_0$ such that $\mathbf{w}^\star(t)$ is a $C^1$ function of $t$ and satisfies the following autonomous system*

$$\dot{\mathbf{w}}^\star(t) \triangleq \Upsilon(\mathbf{w}^\star, z) = - \sum_{k \in I_{L_{n+1}}^a(\mathbf{w}^\star)} \hat{\theta}_{L_{n+1}}^k(\mathbf{w}^\star) \xi(z) \cdot \mathbf{P}(\mathbf{w}^\star | z) \left[ \partial_z \nabla D_{L_{n+1}}^k(y_{n+1}(z), \eta_{\mathbf{w}^\star}(\mathbf{x}_{n+1})) \right], \tag{13}$$

*where $\mathbf{P}(\mathbf{w}^\star | z)$ is obtained from the definition in (11), and the $\hat{\theta}_{L_i}$, $\hat{\theta}_{\Omega_j}$ can be solved from (8).*

The (13) offers an explicit gradient flow of the solution path in underlying optimization space. Solving this ODE numerically would recover all the optimal parametric solutions, which provide the conformity scores and one can obtain the required $p$-values (2) that used for exact conformal prediction via (3).

**Theorem 4.** *The optimality solution $(\mathbf{w}^\star(t), z(t))$ has a unique trajectory for $t \in (0, \mathbb{T})$. The $\mathbf{w}^\star(t)$ is continuous if selections $D_{L_i}^k(\cdot)$, $D_{\Omega_j}^k(\cdot)$ are convex. Furthermore, if gradients of constraints $\{\nabla g_i(\mathbf{w}) : \nabla g_i(\mathbf{w}) = \mathbf{0}\} \cup \{\nabla h_j(\mathbf{w}) : \nabla h_j(\mathbf{w}) = \mathbf{0}\}$ are affinely independent at the solution $\mathbf{w}^\star(z)$ over an open neighborhood of $z$, then the coefficient paths $\hat{\theta}_{g_i}$, $\hat{\theta}_{h_j}$ are unique and continuous at $z$.*

**Theorem 5.** *On a optimality path with set configuration (10), the coefficients for constraints satisfies*

$$\boldsymbol{r}_{\mathcal{Z}} \triangleq \left[ \begin{array}{c} \hat{\theta}_{\mathcal{Z}_E}(\mathbf{w}^\star) \\ \hat{\theta}_{\mathcal{Z}_I}(\mathbf{w}^\star) \end{array} \right] = -\mathbf{Q}(\mathbf{w}^\star) \left[ \frac{1}{\rho} \mathbf{D}'(\mathbf{w}^\star) + \mathbf{u}_{\bar{\mathcal{Z}}}^\top \right], \tag{14}$$

---

[2]In principle a similar algorithm can be developed for the general convex constraint where the $h_j$ are relaxed to convex, but that is beyond the scope of current paper.

*where $\mathbf{Q}(\mathbf{w}^\star)$ is from (11), and*

$$\mathbf{u}_{\bar{\mathcal{Z}}}^\top := -\sum_{i \in \mathcal{N}_E} \mathbf{v}_i + \sum_{i \in \mathcal{P}_E} \mathbf{v}_i + \sum_{j \in \mathcal{P}_I} \boldsymbol{\omega}_j.$$

Although Theorem 3, 4, and 5 are highly technical and may difficult to grasp on first glance, they lay the groundwork for practical application, as illustrated later in Section 4. Specifically, Theorem 4 ensures continuity, which is fundamental for the numerical ODE solvers. Theorem 5, on the other hand, provides a rule for handling constraints.

**Theorem 6.** *Suppose that $D_{L_i}^k(\cdot)$ is $\mu$-strongly convex for $\mu \geq 0$, $D_{\Omega_j}^k(\cdot)$ is $\sigma$-strongly convex for $\sigma > 0$, and $\mathbf{P}(\mathbf{w}^\star|z)$, $\partial_z D_{L_{n+1}}^k(\cdot)$, $\partial_z \nabla D_{L_{n+1}}^k(\cdot)$ are all locally $\ell$-Lipschitz continuous. Suppose further that $\xi(\cdot)$ is Lipschitz continuous on $[z_{\min}, z_{\max}]$ and satisfies $|\xi(z)| \leq \bar{C}$ for all $z \in [z_{\min}, z_{\max}]$. Then, it holds that $\Upsilon(\cdot, \cdot)$ defined in (13) is uniformly $\ell_\Upsilon$-Lipschitz continuous with $\ell_\Upsilon = \bar{C}\ell^2 + \frac{2\bar{C}\ell}{(n+1)\mu + \sum_{j=1}^m \lambda_j \sigma}$ for any $z \in [z_{\min}, z_{\max}]$ when the active selections $I_L$, $I_\Omega$ are fixed.*

The essence of Theorem 6 lies in its suggestion that the dynamics of $\mathbf{w}^\star(t)$ is piecewise continuous, *i.e.*, $\mathbf{w}^\star(t)$ maintains smoothness between two adjacent kinks. By considering specific choices of $\xi(\cdot)$ and $y_{n+1}(z)$, our system (13) generalizes some previously studied methodologies in fast conformalization. First, consider the scenario with an equally spaced discretization of the interval $[0, \mathbb{T}]$, namely $t_k = k \cdot h'$ for some fixed step-size $h' > 0$. Thus, the sequence $z_k := z(t_k)$ is approximately given by $z_{k+1} \approx z_k + h' \cdot \xi(z_k)$. Intuitively, the choice of $\xi(\cdot)$ controls the dynamic of $z(\cdot)$ and generalizes some previously considered sequences $\{z_k\}$ for the problem (5). For example, letting $\xi(z) := 1$ we recover the arithmetic sequence in [12] and letting $\xi(z) := -z$ we recover the geometric sequence in [17].

## 4 Fast conformalization algorithm

The complete algorithm on the fast exact conformalization is outlined in the plate referred as Algorithm 1.

---

**Algorithm 1** Fast Exact Conformalization Algorithm

---

**Input:** Training data $\{\mathbf{x}_i, y_i\}_{i=1}^n$, new covariate $\mathbf{x}_{n+1}$, range $[z_{\min}, z_{\max}]$, initial solution $\mathbf{w}_0^\star$, regularization strength $\{\lambda_j\}_{j=1}^m$, miscoverage level $\alpha \in (0, 1)$.
1: \\ Full Path Generation
2: $z \leftarrow 0$, set $\mathcal{N}_E$, $\mathcal{P}_E$, $\mathcal{P}_I$ and all $I_L$, $I_\Omega$ by $\mathbf{w}_0^\star$.
3: **while** $0 \leq z \leq z_{\max}$ **do**
4:   **while** partitions $\mathcal{N}_E$, $\mathcal{P}_E$, $\mathcal{P}_I$, $I_L$, $I_\Omega$ are met **do**
5:     Calculate $\tilde{\mathbf{H}}(\mathbf{w}^\star|z)$, $\boldsymbol{r}_{\mathcal{Z}}$ as in (12), (14).
6:     Solve ODE system (13).
7:   **end while**
8:   Update $\mathcal{N}_E$, $\mathcal{P}_E$, $\mathcal{P}_I$, $I_L$, $I_\Omega$ by index violator(s).
9: **end while**
10: $z \leftarrow 0$.
11: **while** $z_{\min} \leq z \leq 0$ **do**
12:   Repeat the above procedure analogously for negative values of $z$, obtaining $\{\mathbf{w}^\star(z) : z_{\min} \leq z \leq 0\}$.
13: **end while**
14: \\ Conformal Set Generation
15: **for** $i = 1$ to $n + 1$ **do**
16:   Calculate conformity score path $\mathcal{A}_i$ for $i$-th sample.
17: **end for**
18: Calculate the path of $\hat{p}_{y_{n+1}}$ by (2).
19: $\Gamma(\mathbf{x}_{n+1}) \leftarrow \{y_{n+1} : \hat{p}_{y_{n+1}} \geq \alpha\}$.
**Output:** Conformal prediction set $\Gamma(\mathbf{x}_{n+1})$.

---

Table 2: Numerical results for average empirical coverage, average length of the conformal prediction set, and the *total* number of kinks observed. One standard error is given in the parenthesis following the average number.

| Dataset | Model (Parameter) | Grid1/Grid2 | | SCP | | Exact | | # Kinks |
|---|---|---|---|---|---|---|---|---|
| | | Coverage | Length | Coverage | Length | Coverage | Length | |
| fried | NLS(\) | 0.81(0.003) | 5.82(0.53) | 0.81(0.007) | 5.55(0.94) | 0.80(0.009) | 5.30(0.72) | 6 |
| cadata | NLS(\) | 0.82(0.007) | 5.54(0.12) | 0.82(0.007) | 5.63(0.94) | 0.80(0.003) | 5.35(0.02) | 2 |
| delta | NLS(\) | 0.81(0.008) | 5.63(0.16) | 0.81(0.002) | 5.79(0.53) | 0.80(0.002) | 5.26(0.27) | 5 |
| cadata | GFM($\lambda_1 = 0.03$) | 0.82(0.001) | 11.64(0.13) | 0.81(0.004) | 11.38(0.36) | 0.80(0.003) | 11.34(0.27) | 7 |
| cadata | GFM($\lambda_1 = 0.02$) | 0.83(0.008) | 10.72(0.91) | 0.81(0.004) | 10.39(0.97) | 0.80(0.001) | 10.13(0.21) | 5 |
| elevator | GFM($\lambda_1 = 0.03$) | 0.83(0.003) | 11.09(0.58) | 0.81(0.007) | 11.34(0.97) | 0.80(0.002) | 10.68(0.81) | 9 |
| fried | IGR($\lambda_1 = 0.1, \lambda_2 = 0.02$) | 0.81(0.002) | 19.14(0.78) | 0.81(0.001) | 19.69(0.96) | 0.81(0.002) | 19.05(0.19) | 12 |
| cadata | IGR($\lambda_1 = 0.1, \lambda_2 = 0.02$) | 0.82(0.003) | 19.99(0.33) | 0.80(0.002) | 19.16(0.29) | 0.80(0.002) | 19.14(0.82) | 13 |
| elevator | IGR($\lambda_1 = 0.2, \lambda_2 = 0.05$) | 0.82(0.003) | 19.99(0.85) | 0.81(0.001) | 19.53(0.53) | 0.81(0.005) | 19.48(0.75) | 7 |

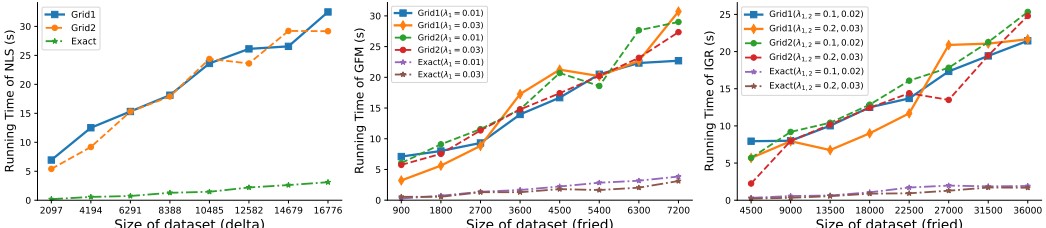

Figure 2: The running time under different sizes of datasets.

## 4.1 On solving (11)

To avoid the direct computation of matrix inversion (11), our practical algorithm involves sweep operator [22, 23]. Suppose $\mathbf{A}$ is an $p \times p$ symmetric matrix, sweeping on the $k$-th diagonal entry $A_{kk} \neq 0$ of $\mathbf{A}$ results in a matrix $\widehat{\mathbf{A}}$ with entries

$$\hat{A}_{kk} = -\frac{1}{A_{kk}}, \quad \hat{A}_{ik} = \frac{A_{ik}}{A_{kk}} \, (i \neq k), \quad \hat{A}_{kj} = \frac{A_{kj}}{A_{kk}} \, (j \neq k), \quad \hat{A}_{ij} = A_{ij} - \frac{A_{ik}A_{kj}}{A_{kk}} \, (i, j \neq k). \tag{15}$$

Since the sweeping (15) preserves symmetry, all operations can be performed solely on either the lower or upper-triangular part of $\mathbf{A}$ to ease the computational burden [24]. To begin with, we initiate with a sweeping tableau as $\left[ \begin{array}{c|c} \tilde{\mathbf{H}}(\mathbf{w}^\star | z) & * \\ \hline \mathbf{U}_{\mathcal{Z}} & \mathbf{O} \end{array} \right]$, and further sweeping of diagonal entries of block $\tilde{\mathbf{H}}$ yields $\mathbf{M} \triangleq \left[ \begin{array}{c|c} \mathbf{M}_{11} & * \\ \hline \mathbf{M}_{21} & \mathbf{M}_{22} \end{array} \right]$. Then we reinitialize our new tableau in the form of $\left[ \begin{array}{c|c} -\mathbf{M}_{22} & \mathbf{M}_{21} \\ \hline * & -\mathbf{M}_{11} \end{array} \right]$, and further sweeping of diagonal entries of block $\mathbf{M}_{22}$ makes $\left[ \begin{array}{c|c} \mathbf{R} & \mathbf{Q}^\top(\mathbf{w}^\star | z) \\ \hline * & \mathbf{P}(\mathbf{w}^\star | z) \end{array} \right]$. Compared to direct inversion, it also decreases a $\mathcal{O}(p^2 + n_{\mathcal{Z}}^2)$ storage space.

## 4.2 Efficiency

Algorithm 1 shows a very favourable behavior empirically, and converges remarkably faster than the standard grid-search type algorithm. We argue that this observation is actually quite natural. Indeed, our algorithm can follow the ground truth solution path, and the numerical integration process is fully deterministic, which avoids large fluctuations between the iteration steps like stochastic gradient descent. Therefore, the solving process of Algorithm 1 exhibits a more stable character being completely *deterministic* and has no extensive loops, which explains the much faster convergences observed in practice. In contrast, the standard grid-search type method would cost $N$-times the original batch iterative algorithms, where $N$ is the number of grid points.

## 5 Numerical experiments

We provide experimental results on real-world benchmarks to validate our derived algorithm. All experiments presented in this study were conducted on a workstation running the Ubuntu 18.04 operating system, equipped with Intel Xeon Gold 5218R CPU×64 and 60.9 GB of RAM. We integrate a system of ordinary differential equations using lsoda from the FORTRAN library, where an

interface for `SciPy` is available using the `odepack`. The concrete parameter settings of ODE solver are shown in the Table 3, wherein the numerical solver exploit the Runge-Kutta method of order 4 or 5. The parameterizers are set to $y_{n+1}(z) = 4z, \xi(z) = 1$, respectively.

Table 3: List of the key parameters in the numerical solver.

| Parameters | Descriptions | Values |
|---|---|---|
| rtol | allowed relative error in the solution | 1e-6 |
| atol | allowed absolute error in the solution | 1.49e-8 |
| tcrit | vector of critical points | *set by known / explicit kinks* |
| h0 | initial step size for the integration | $0.02*(z_{max} - z_{min})$ |
| hmax | maximum absolute step size allowed | $0.1*(z_{max} - z_{min})$ |
| hmin | minimum absolute step size allowed | 1e-7 |
| mxstep | maximum number of steps allowed for each point | 400 |
| mxordn | maximum order to be allowed for the non-stiff method | 12 |
| mxords | maximum order to be allowed for the stiff method. | 5 |
| ixpr | extra printing at method switches | True |
| mxhnil | maximum number of messages printed | 15 |
| tfirst | the required order of the first two arguments | True |
| full_output | return a dictionary of optional outputs | True |

**Model**    For evaluation, here we employ 3 specific forms of (1), *i.e.*, Nonnegative Least Squares (`NLS`) [25], Graph-guided Fused Model (`GFM`) [26], and Inverse Gaussian Regression (`IGR`) [27].

**Conformalization**    A conformal prediction set with target coverage level $0.8$ ($\alpha = 0.2$) is calculated for each sample in testing set using each of the 4 methods, *i.e.*, the standard grid-point evaluation method (`Grid1`) [1], grid-point method with warm-restart strategy (`Grid2`) [28], the split conformal prediction method (`SCP`) [4], and our exact conformalization method in Algorithm 1 (`Exact`). We use the conformity score function $\mathcal{A}_i = -|y_i - \eta_{\mathbf{w}^\star}(\mathbf{x}_i)|$. Conventionally, the interval $[y_{n+1}^{\min}, y_{n+1}^{\max}]$ (part of the input) can be chosen simply as $[y_{[1]}, y_{[n]}]$, where $y_{[1]} \leq y_{[2]} \leq \cdots \leq y_{[n]}$ are the order statistics of the response variable. In experiments we set the search range even more conservatively, enlarging the sample range by 50% of length $[y_{n+1}^{\min}, y_{n+1}^{\max}] := [y_{[1]} - 0.25(y_{[n]} - y_{[1]}), y_{[n]} + 0.25(y_{[n]} - y_{[1]})]$.

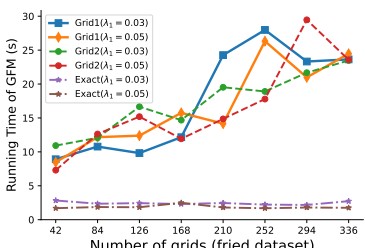

Figure 3: The training time against grid numbers.

**Dataset**    Our experiments were conducted using real-world datasets. We employ real-world datasets from OpenML [29] and UCI repository [30] in simulations. We randomly partition the dataset into training set, testing set, and calibration set (used in `SCP`) with 70%, 10%, and 20% of the total samples. To facilitate optimization, we have standardized the entire original dataset by removing the mean and scaling to unit variance for the features, and adjusting the mean of all labels to 0.

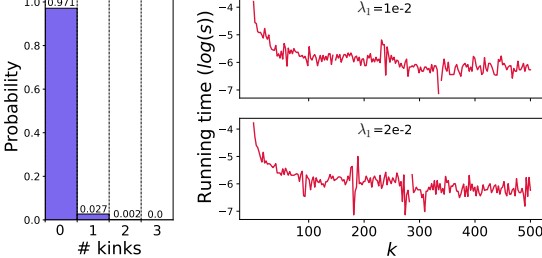

Figure 4: The histogram of kink numbers and the running time of Algorithm 1 against $k$, where $k$ lives in $y_{n+1} := k \cdot z (k \in \mathbb{N})$.

**Setup**    Our central claim is twofold, encompassing both accuracy and efficiency. We first report the average empirical coverage, average length of the prediction set in Table 2. Regarding running efficiency, we present average training time per dataset in Figure 2 while varying the scale of training set. In Figure 3, we compare the training times when different grid numbers are used in `Grid1` and `Grid2`. We further plot the histogram of kink numbers, and the running time against various $y_{n+1}(\cdot)$ in Figure 4.

**Results & Analysis**    From Table 2 we observe that all these methods provide valid and nearly perfect coverage. The grid and exact method give similar lengths, where the slight difference is due

to the rounding between neighboring grid points. The SCP produces wider intervals due to a less efficient use of data. Given Figure 2, our exact method is much faster than the baselines, with same solid performance. We can also learn from the Figure 3 that the time of Algorithm 1 still compares favorably against the grid-search type method, even when the grid is sparse. Figure 4 found that there exists tiny number of kinks in majority runnings, which offers hope for the future expansion of our algorithm, and indicates that the choice of $y_{n+1}$ can make a difference in efficiency, as it will determine the solving interval and the total query times of gradient.

# 6   Conclusion

In this work, we present a unified framework and an elaborate algorithm with statistical analysis for fast exact conformalization regarding generalized parametric estimation. We illustrate the strong and competitive performance of proposed methods in a series of benchmarks.

In future work, a potential direction is to consider scenarios where labels are multidimensional, such as multi-task learning, in which the label space $\mathcal{Y}$ would be indexed by multiple independent parameters $(z_1, \ldots, z_K)$. Under such conditions, the homotopy solution path would extend to a solution surface, and our ODE system could be reformulated as a corresponding system of partial differential equations. Additionally, compared to some previous work, Algorithm 1 offers increased speed but at the cost of higher memory requirements. It necessitates storing all training samples throughout the process for gradient queries. We believe it would be interesting to use recent advancements like the Kronecker-factored approximate method to potentially enhance the memory scalability.

# Acknowledgement

I would like to thank the anonymous reviewers for their valuable comments and suggestions to improve the presentation of this paper.

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
