# OpenReview forum: "Generalized Fast Exact Conformalization"
_NeurIPS.cc/2024/Conference — NeurIPS 2024 poster_

### Official Review · Reviewer_jxtf · 2024-07-04

**Soundness:** 3
**Presentation:** 2
**Contribution:** 2
**Rating:** 6
**Confidence:** 3

**Summary:**

The authors study the *solution path* of constrained optimization problems when one training label varies in ${\mathbb R}$. In the Conformal Prediction framework, this is equivalent to studying the output of a Full-CP algorithm. The paper extends the approach to optimization problems with general constraints and a piece-wise differentiable objective.

**Strengths:**

- Characterizing the solution path of a series of smoothly varying optimization problems is relevant, even beyond the CP framework.
- The Lagrangian reparametrization approach is interesting.
- The proposed method seems to be more efficient than Split CP.

**Weaknesses:**

- It would be helpful to clarify from the start what "opening the black box" means, e.g. by listing the assumptions needed to obtain the solution path efficiently.
- The authors should justify better why they need
i) general constraints and
ii) piece-wise differentiable objectives.
A warm-up section describing the unconstrained case would also help. Has the unconstrained and globally differentiable case been considered elsewhere?
- The method seems to apply in a convex neighbor of the optimum. Assuming the algorithm is initialized there, the optimization problem is quadratic. The authors should comment on the difference between their method and the standard full-CP approach in the presence of several local optima.

**Questions:**

- In the abstract, you mention the computational burden of full-CP algorithms. What about Split-CP? Does "strong assumptions" mean data inefficiency?
- Is the solution path unique? What happens for overparametrized models?
- In what sense the prediction sets of [18] and [19] do not have statistical guarantees? Can't one combine the upper bound on the estimation error and standard CP validity?
- Is the formalism developed in Section 3 only needed because of the general constraints and piece-wise assumption in  Equation 1?
- Intuitively, what does produce a kink in $w_*(t)$?
- Why do you fix $\alpha=0.8$?
- In Algorithm 1, does $z$ need to be discretized?
- Why Split CP does not appear in Figures 3 and 4?
- Why is the standard approach, i.e. ``Grid1/Grid2`` in Table 2, outperformed by Split CP? Normally full-CP is expected to be more efficient.

**Limitations:**

The authors do not discuss the limitations of their work. They may have added a few lines about the applicability of their assumption and the limited choice of predictors in the experiments.

---

> ### Author Rebuttal · Authors · 2024-08-07
>
> # Many thanks for appreciating our work!
> Dear reviewer jxtf,
>
> We deeply appreciate your commitment in reviewing our paper and your encouraging words of support for our work! In the following, we will provide a comprehensive response to your review comments.
>
> ---
> > clarify what "opening the black box" means
>
> Please refer to general rebuttal, section ②. Thank you!
> > should justify better why they need i) constraints ii) piece-wise differentiable
>
> This is primarily **to achieve the versatility of the objective**. Nowadays many learning tasks essentially involve unconstrained optimization, but it's well known that many loss functions are not globally differentiable, so we believe this extension is natural. Moreover, constraints are also not rare in real-world tasks, e.g., ordinal support vector machines used for classification and regression [1], linear programming in operations research [2], and certain fairness constraints in fairness classifications [3], all of which can be accommodated within our Eq. (1). Several models in experiments also validate this, as current work cannot handle their conformalizations.
>
> Following your constructive suggestion, **we added relevant examples and further motivation** in Section 2.
> > Has the unconstrained and globally differentiable considered elsewhere?
>
> All baselines prior to our work have studied the unconstrained case (Table 1), and mostly require the objective function to be globally differentiable. For research on solution paths that unrelated to conformalization, please refer to Appendix F.1.
> > The method seems to apply in a convex neighbor of the optimum
>
> We believe that homotopy continuation does not necessarily occur within the convex neighborhood (even if the selection function is strongly convex), but is generally within a closed or bounded neighbor of the optimum. More theoretical properties will be explored in future work.
> > The authors should comment on the difference ... in the presence of several local optima
>
> Standard full-CP uses batch-training solvers (e.g. various SGD), making it difficult to control their optimization behavior. Whether the solver can escape saddle points depends on its inherent properties and user parameters. On our algorithm, please see next question.
>
> We added further explanations based on your comments.
> > Is the solution path unique? What happens for overparametrized models?
>
> Computationally, the Picard–Lindelöf theorem proves that the solution of numerical ODE is unique. Geometrically, if regularity conditions are met, then points of the path (with non-vanishing KKT multipliers) is locally the projection of a higher-dimensional smooth manifold onto optimization space, hence it's not unique but our algorithm can indeed follow a path of stationarity points.
>
> Overparametrized case: please see Appendix F.4.
> > computational burden: What about Split-CP? Does "strong assumptions" mean data inefficiency?
>
> (Characters limitation) please see response to Reviewer 4Aeq. Thanks!
>
> It means that previous methods cannot be extended and are not sufficiently general.
> > In what sense the [18, 19] do not have statistical guarantees? Can't one combine the ...?
>
> Thanks for valuable feedback! This thought is reasonable, and aligns with one of the motivations behind these works. We have revised our writing to indicate that there are no optimal statistical guarantees, as their solutions for minimization are indeed not exact.
> > Is the formalism developed in Section 3 only needed because of the general constraints and piece-wise assumption?
>
> Yes, our framework would be simplified (but less useful) without the above settings.
> > what does produce a kink in $\mathbf{w}^\star(t)$?
>
> Kinks (or non-smooth transition points) on the path are caused by non-differentiable points, including boundary constraints hitting or the non-differentiable points within the loss / regularizer itself (e.g., the $\ell_1$ norm at $0$). Appendix C.3 provides an intuitive illustration of kinks in a 2-dimensional optimization space.
> > Why fix $\alpha=0.8$?
>
> This is only needed for numerical simulations. Our algorithm primarily focuses on *full path generation*, while $\alpha$ is used in *conformal set generation* (the second stage). Like the baselines, we use standard way to compute the conformal set, so **the setting of $\alpha$ has no impact on our core algorithm**. We observed that most related works set $\alpha$ around 0.1 or 0.2, so our setup is reasonable. Let us know if you have specific concerns.
>
> Another insight from your question is that we could consider $\alpha$ as a *variable* to study the underlying relationship between the resulted conformal set and $\alpha$, but this beyond the current scope.
> > does $z$ need to be discretized?
>
> The need for discretizing $z$ depends on the property of the label space $\mathcal{Y}$. In regression, since $y_{n+1}$ is continuous, there's no need to discretize $z$. In classification problems, $z$ is generally discrete for computational purposes, as we focus on the discrete $y_{n+1}$ of interest.
> > Why SCP does not appear in Figures 3, 4?
>
> (Characters limitation) please see response to Reviewer ji5B. Thanks!
> > Why is the standard approach outperformed by Split CP?
>
> Indeed, the SCP results are better in a few lines. That's may due to our choice of grid points not being overly dense to ensure fairness in comparisons. That's also related to the data distribution and calibration set partitioning in SCP.
> > The authors do not discuss the limitations
>
> Technical limitations are described in Appendices F.3 and F.4.
>
> ---
> **References:**
>
> [1] "Support vector machines–an introduction." Springer, 2005
>
> [2] "Linear programming and its applications." Springer, 2007
>
> [3] "Fairness constraints: Mechanisms for fair classification." AISTATS 2017
>
> **We thank the Reviewer jxtf again for your insightful feedback and strong support of our submission! If you have any remaining concerns or further inquiries, please do not hesitate to let us know.**

---

### Official Review · Reviewer_KUWB · 2024-07-08

**Soundness:** 2
**Presentation:** 1
**Contribution:** 2
**Rating:** 3
**Confidence:** 1

**Summary:**

This paper proposes a new method to accelerate the computation of full conformal prediction sets, a task that traditionally requires a computationally intensive grid-search approach. The proposed method aims to streamline this process, potentially reducing the need to refit the predictive model for each test point and each possible outcome value. However, the paper suffers from poor writing, heavy mathematical notation, and an overly dense presentation, while also employing tricks like small font sizes in figures and text wrapping to extend beyond the standard page limits. Due to these significant issues, I believe that a thorough review of this paper is not feasible at this time. I would consider this a 'desk rejection' from my perspective. I hope the authors will not be offended or discouraged, as this is not a judgment on the quality of their research, which I was unable to carefully assess, but rather an invitation to improve the presentation to facilitate the review process.

**Strengths:**

- This paper studies an important problem in conformal inference, namely to speed up computations for full-conformal prediction.
- The proposed method could potentially reduce the computational burden associated with conventional grid-search approaches.

**Weaknesses:**

- The paper is not very well written, it's very dense, and difficult to understand, even for those familiar with the broader field.
- The paper is too long and only fits within the page limits due to some "tricks" such as using very small font sizes in figures and wrapping text around figures.

**Questions:**

- Could you try to enhance the writing quality and clarity of the paper to make it more accessible to a broader audience?
- Would you consider shortening the paper, or alternatively, submitting it to a venue that accommodates longer articles? This paper might turn out to be much easier to understand in an extended version. My philosophy is that if the content cannot be effectively conveyed within 9 pages, it is better suited for a different format. Forcing it to fit within this limit does a disservice to the readers by compromising clarity and comprehensiveness.

**Limitations:**

The primary limitation of this paper is its poor writing quality and extremely dense presentation, making it difficult to follow and understand. In its current form, the potential audience of this paper might at best be limited to a very narrow group of readers, which unfortunately does not include this reviewer.

---

> ### Author Rebuttal · Authors · 2024-08-07
>
> # Thankful for your constructive feedback!
> Dear reviewer KUWB,
>
> We are very grateful for the effort you have put into reviewing our work and for recognizing the importance of our research problem. We have addressed and acted upon your main criticisms, and we are looking forward to further interaction with you.
>
> ---
> ## Ⅰ. Text Wrapping & Figure Size
>
> > employing tricks like small font sizes in figures and text wrapping to extend beyond the standard page limits
>
> >  due to some "tricks" such as using very small font sizes in figures and wrapping text
>
> - Thank you for your feedback. Firstly, we would like to politely point out that *text wrapping* and *figure scaling* are ***not*** tricks we specifically employed in this paper. **Such practices are quite common at NeurIPS and other similar venues like ICML / ICLR**.
>   - *e.g.*, P7 in [1], P6-P8 in [2], P8 in [3], P5-P7 in [4], etc.
>
> - We understand your concerns. All the images inserted in this paper, whether experimental results or logical diagrams, are **vector graphics and can be enlarged arbitrarily without distortion**.
>   - Based on your feedback, **we promise to further adjust the layout** by merging some runtime comparison figures with sections in the appendix. This will allow us to **increase the font size of each figure** in the main body.
>
> - Besides, kindly note that **if this paper was accepted, we will have one extra page** in the camera-ready version.
>
> - As per our response to Reviewer ji5B, **we will also move some minor technical content to the appendix or remove it entirely.** For example, some corollaries, Sections 4.1 and 4.2. This will create further space in the main body, reducing its content density.
>
> ## Ⅱ. Clarity & Length
>
> > However, the paper suffers from poor writing
>
> > The primary limitation of this paper is its poor writing quality
>
> Despite the mathematical notation being dense (in some sections), our overall structure should be pretty clear and is well organized, as acknowledged by other reviewers. Our introduction part (including *motivation*, *high-level ideas*, and *related work*) and experimental sections have been well understood by several reviewers. This indicates that **the presentation of our paper, while not perfect, still meets the basic standards of a technical paper**. We are also continuously improving our writing.
>
> > The paper is too long
>
> > the content cannot be effectively conveyed within 9 pages... Forcing it to fit within this limit does a disservice by compromising clarity and comprehensiveness
>
> - We fully understand your worries regarding the length and formatting of the paper. At first, please let us kindly point out that **all content in our appendix merely provides further explanations of the technical results in the main bofy, rather than introducing new arguments.**
> Even if we were to remove the entire appendix (except proofs), the content of our main body still remains entirely self-contained, which has included the core theorems, discussions on assumptions, and statistical descriptions of the novel framework.
>
>   - Any reader familiar with optimization and conformal prediction can follow the algorithmic steps presented in the 9 pages, without reading the appendix. Therefore, we respectfully *disagree* with the viewpoint that the content cannot be effectively conveyed within main body.
>
> - We believe the length and (technical) clarity involve a trade-off.
>   - We could, of course, write several hundred pages listing every definition and detailing every step of derivations, but this would be inappropriate for a formal research paper.
>   - Thus, in addition to the main body, we have included extensive discussions in the appendix, which help readers understand the core framework from various perspectives. **These in-depth discussions and extra experiments aim to enhance the *clarity* and *technical comprehensiveness* of our research findings**, stimulate readers' thoughts on the algorithmic details, and attract researchers with different backgrounds. (as some content like *IVP in differential equation* might be overly fundamental for readers who are already familiar with them)
>
> - In appendix we dedicated **10 pages to study our by-product**. Since the main focus is conformalization, **we can simply delete this part and immediately reduce the paper's length by nearly 20%**.
>
> > Would you consider shortening the paper
>
> **Yes**! Thanks again for your advice.
>
> ## Ⅲ. Community Interest
>
> >  difficult to understand, even for those familiar with the broader field
>
> > audience might at best be limited to a very narrow group of readers
>
> Thank you for your thoughtful comments. Fully understanding the details of theoretical framework indeed requires *a certain level of math maturity*. However, **for those engaged in algorithmic research, referring to the pseudo-code and the explanations provided in the appendix should be sufficient** to apply our new algorithm, and promote its application at a more practical level, even for those outside the ML community.
>
> As we explained, the **contributions of this work encompass both algorithmic and new theoretical advancements**. By NeurIPS guideline, this community spans over optimization theory, statistics, and probabilistic ML. I believe our work will also attract researchers beyond those specializing in conformal prediction.
>
> ---
> **References:**
>
> [1] "Learning to Relax: Setting Solver Parameters Across a Sequence of Linear System Instances." ICLR 2024
>
> [2] "In-context impersonation reveals Large Language Models' strengths and biases." NeurIPS 2023
>
> [3] "Steve-1: A generative model for text-to-behavior in minecraft." NeurIPS 2023
>
> [4] "SaNN: Simple Yet Powerful Simplicial-aware Neural Networks." ICLR 2024
>
> **Meanwhile we would like to highlight our main contributions again in the following.** *We respectfully ask you to read our response and consider stronger support based on the soundness and contribution of this work.* Our sincere thanks once again.

---

> > ### Comment · Reviewer_KUWB · 2024-08-09
> >
> > Dear Authors,
> >
> > Thank you for understanding that my review was not a critique of the content itself but rather of its presentation. As I mentioned, this paper seems ill-suited to fit within the 9-page limit. If the paper were only slightly over the limit, I would not have raised an issue. However, the current issues due to length, density and quality of presentation are significant in my opinion. It's also indicative that, despite their overall positive assessment, other reviewers have also noted that your paper is hard to read.
> >
> > This raises three main concerns:
> >
> > - **Reviewer Burden**: I volunteered to review a 9-page paper, with the expectation that the authors had made every effort to present their work clearly. When the paper is overly long and dense, it becomes difficult to review thoroughly within the time constraints. If I cannot review the paper carefully, I am unable to fully verify its technical correctness or assess its novelty, which are critical factors in recommending it for publication. The fact that other reviewers have given positive scores does not justify altering my independent assessment.
> >
> > - **Suitability for NeurIPS:** If the paper is too lengthy and complex for a reviewer to digest, it is likely that many NeurIPS readers will find it similarly inaccessible. This brings into question whether NeurIPS is the right venue for this work.
> >
> > - **Impact:** If both reviewers and potential readers struggle with the paper's density, it raises the question of why the authors are intent on publishing it in its current form. I strongly recommend that you take my feedback seriously and consider revising the paper to make it more accessible. Alternatively, you might explore a venue that allows for a longer version, where the absence of strict space constraints would enable you to present your work more clearly. If your goal is to make an impact, enhancing the paper's readability is in your best interest. Not every paper can—or should—be condensed into 9 pages.
> >
> > In conclusion, I suggest you consider my feedback carefully. I have assigned a low confidence score to my review, acknowledging that I was unable to thoroughly assess the paper. This is the most responsible course of action I can take under the circumstances.

---

> ### Author Response · Authors · 2024-08-07
> **Highlighting Our Contributions**
>
> We would like to highlight our main contributions again in the following. *We would be very happy to engage with reviewers if they have any doubt or confusion.* Thank you.
>
> ---
> 1. **We are the first to achieve a fast exact conformalization for generalized parametric estimation**, with accuracy comparable to the existing grid-search type method. This process is underpinned by rigorous theoretical analysis, demonstrating that the **Algorithm 1 output is theoretically equivalent to ground truth solution path**. Please see our general response, and for further details, refer to Section 3.3 and Appendix F.2. Traditionally, the only way to construct a conformal prediction set has been limited to loop over the label space $\mathcal{Y}$, which discretize the interval of interest and subsequently solve a sequence of individual optimization subproblems [1].
> Our contribution thus **addresses a long-standing gap in the field**, significant for both the conformal prediction and optimization communities.
>
> ---
> 2. **Our algorithm is applicable to any generalized parametric estimation, provided mild assumptions are met** (see Assumptions 1, 2). This general applicability was previously unattainable in literature, which focused on specific ML models. For example, (Lei, 2019) [13] provided theoretical insights for the $\ell_1$-regularized quadratic loss (Lasso), *i.e.*, let $t_0=0$, $J_0 = \{j : \hat{\beta}(0) \neq 0\}$, the piecewise linearity of model solution $ \hat{\beta}$ is
> $$\begin{align*}
> \eta(k) &=\frac{n^{-1}\sum\_{j=1}^{J\_k} x\_{n+1,jk}}{1+n^{-1} x\_{n+1,J\_k} \sum\_{j=1}^{J\_k^{-1}} x\_{n+1,jk}}, \\\\
> \gamma(k) &=\frac{x\_{n+1,J\_k^c} - \sum\_{j=k-1}^{J\_k^c} \sum\_{j=1}^{J\_k^{-1}} x\_{n+1,jk}}{1+n^{-1} x\_{n+1,J\_k} \sum\_{j=1}^{J\_k^{-1}} x\_{n+1,jk}}, \\\\
> \hat{\beta}\_{J\_k}(t) &= \hat{\beta}\_{J\_k}(t\_k)+\eta(k)(t - t\_k) \quad \forall t \in [t_k, t_{k+1}],\hat{\beta}\_{J\_k^c}(t)=0, \\\\
> v\_{J\_k^c}(t) &=v\_{J\_k^c}(t\_k)+\gamma(k)(t-t\_k) \quad \forall t \in [t_k, t_{k+1}],
> \end{align*}$$
>
> but benefits to researchers using other forms of parametric models were pretty limited, since **formulating similar rules for more complicated parametric estimation could be theoretically challenging**. Our method requires only weak assumptions, enabling exact conformalization for a range of significant and classic models like **Non-negative Least Squares and Inverse Gaussian Regression for the first time**. We offer a unified framework for this pivotal domain of generalized parametric estimation, which is indeed one of our primary contributions.
>
> ---
> 3. **Our framework is straightforward to implement and computationally efficient**. With theoretical analysis, we present explicit expressions for the gradient flows of reparameterized optimization problem, **homogenized and aligned with standard forms used by mainstream numerical libraries**, simplifying programming efforts. We pointed out the complexity of algorithmic implementation in our response to Reviewer ji5B (also see Section 4.3). Computationally, our solver adaptively selects step sizes within the solution interval to capture all events / kinks and can swiftly adjust the active set through preset conditions. This feature is wll-supported by many solver libraries. Our algorithm requires no extensive iterations for traversing latent parameters, contrasting sharply with conventional grid-search type techniques.
>
> ---
> 4. **This research reveals the dynamics of solution and the essential structure of optimization paths for conformalized parametric estimation (Section 3.3), which is of independent intellectual interest** and also bears significant implications for machine learning.
> We use dynamics to reveal how estimation depends on latent parameters, which coincides with recent work in the optimization community [3,4].
> The idea of combining differential equation systems with optimization paths adds a significant tool to community's toolbox, and provides a bridging interface between machine learning and applied mathematics.
>
> ---
> 5. As a by-product, **we introduce an exact online label-varying algorithm**. Our analysis indicates that it can adapt label changes effectively, where the updated online solution **is equivalent to retraining from scratch** on new labels using standard batch approaches.
> We also conducted numerical evaluations in simulated label-varying environments to demonstrate its accuracy and efficiency.
>
> ---
>
> **References:**
>
> [1] Papadopoulos, Harris, et al. "Regression conformal prediction with nearest neighbours." Journal of Artificial Intelligence Research (2011).
>
> [2] Lei, Jing. "Fast exact conformalization of the lasso using piecewise linear homotopy." Biometrika (2019).
>
> [3] Lin, Xi, et al. "Continuation path learning for homotopy optimization." ICML 2023.
>
> [4] Negri, Marcello Massimo, et al. "Conditional Matrix Flows for Gaussian Graphical Models." NeurIPS 2023.

---

> ### Author Response · Authors · 2024-08-09
> **Many thanks for your kind follow up!**
>
> Dear reviewer KUWB,
>
> Thank you for your prompt response along with your kind understanding. We deeply appreciate your candid feedback regarding the low confidence score as well as your responsible actions. Please allow us to address any remaining concerns that you may have:
>
>
> - **Reviewer Burden.**
> At the beginning, we greatly appreciate your time and effort in evaluating our paper. We sincerely accept your criticism and understand that this paper has caused additional devotion for some reviewers, including yourself, but this was never our intention. Our research focuses on the conformalization of generalized estimation, so we aimed for it to be comprehensive (i.e., broadly applicable) and to provide in-depth technical insights. As stated in rebuttal, the main body of this paper is completely self-contained. Our 9-page main text sufficiently describes our motivation, related arts, and presents our theoretical findings as well as the algorithmic steps. The sections in appendix are only necessary when one need to learn the experimental details, or verify our theoretical derivations step-by-step.
>
>
> - **Clarity.**
> We understand and respect your concerns regarding the paper's density and presentation. We place great importance on your proposed suggestions.
>
>   1. By removing some less critical technical paragraphs (e.g., Section 4.1, 4.4) and minor results (e.g., Corollary 1, Theorem 7), we have gained additional space in the main body and adjusted the equations and figures to reduce content density.
>
>   2. We have carefully revised the vague sentences in main body, incorporating feedback from all reviewers.
>
>   3. We have also shortened the overall length of the paper based on your valuable feedback; specific details are listed below.
>
>    - Although the notation in some sections is dense, as mentioned in our rebuttal, we believe that our overall textual writing meets the minimum required standards.
>
>    - We already had a detailed symbol table in appendix, allowing readers to quickly query the meanings of the math notations.
>
>
> - **Audience.**
> Mathematically, some reviewers explicitly pointed out that our work is hard to follow, but several other reviewers’ comments and feedback reflect their deep understanding of the paper. In a summary, for those with a certain level of mathematical maturity, our paper offers theoretical tools and a statistical framework. For researchers in related fields, it provides practical algorithms and a high-level theoretical narrative. Our work essentially bridges different domains, while the detailed explanations in the appendix lower the barrier to understanding, aiming to engage a broader audience.
>
>
> - **NeurIPS Suitability.**
> While we do understand your concerns about the paper’s suitability for NeurIPS, detailed appendices are not uncommon in conference presentations. We would like to respectfully point out that our current paper length is relatively long, but not excessively so. In the next response box, we will provide some examples from last year's NeurIPS to illustrate this point.
>
>
>
> We acknowledge and respect your decision regarding the score assigned to our submission. Regardless of whether this paper is ultimately accepted, we are committed to continuously improving its clarity and readability. On this point, we believe that we share a common pursuit. We once again express our gratitude for the constructive conversation with reviewer KUWB.
>
> Sincerely,
>
> Authors
>
> ---
>
> $ $
>
>
> ###  Length reduction
>
> Based on the feedback from reviewers, we have made the following major changes to our appendix. **The final effective length of paper will be reduced from the current 51 pages to 36 pages**:
>
> - We removed many minor references, reducing the 6-page reference list to only 4 pages. Additionally, we deleted the table of contents (*page 16*).
>
> - We significantly reduced the additional numerical results (*pages 41-43*) to less than 1 page.
>
> - We have shortened Appendix F: Additional Discussions (*pages 43-46*) to only 1 page.
>
> - Given that the focus of this paper is on conformal prediction, and only one of the five reviewers mentioned our by-product (i.e. Algorithm 2), we will remove discussions and simulations related to this by-product that are not directly relevant to our key focus. This includes the theoretical analysis (*pages 34-35*), examples (*pages 38-39*), and discussions and experiments (*pages 47-51*).

---

> ### Author Response · Authors · 2024-08-09
> **Supplement: relatively long papers**
>
> From last year’s NeurIPS conference proceeding, we randomly picked some of the relatively longer papers. To avoid the bias and to maintain convincingness, all the papers listed below are only selected from the *Spotlight* or  *Oral* batches.
>
> ----
>
> In some papers, authors provide additional theoretical insights and discussions in their appendix. Examples include:
>
>
> [1] "Survival instinct in offline reinforcement learning." NeurIPS 2023. `(59 pages)`
>
>
> [2] "Clifford group equivariant neural networks." NeurIPS 2023. `(69 pages)`
>
>
> [3] "Adversarial training from mean field perspective." NeurIPS 2023. `(54 pages)`
>
>
> [4] "Monarch mixer: A simple sub-quadratic gemm-based architecture." NeurIPS 2023. `(58 pages)`
>
>
> [5] "Transformers as statisticians: Provable in-context learning with in-context algorithm selection." NeurIPS 2023. `(87 pages)`
>
>
> [6] "Bridging RL theory and practice with the effective horizon." NeurIPS 2023. `(55 pages)`
>
>
> [7] "Decentralized randomly distributed multi-agent multi-armed bandit with heterogeneous rewards." NeurIPS 2023. `(57 pages)`
>
>
> [8] "Understanding multi-phase optimization dynamics and rich nonlinear behaviors of relu networks." NeurIPS 2023. `(94 pages)`
>
>
> [9] "Would I have gotten that reward? Long-term credit assignment by counterfactual contribution analysis." NeurIPS 2023. `(51 pages)`
>
> ---
>
> Authors also sometimes supplement the appendix with additional experiments to enhance the credibility of their research, such as:
>
>
> [10] "The goldilocks of pragmatic understanding: Fine-tuning strategy matters for implicature resolution by LLMs." NeurIPS 2023. `(79 pages)`
>
> [11] "Okridge: Scalable optimal $k$-sparse ridge regression." NeurIPS 2023. `(183 pages)`
>
> [12] "Parsel🐍: Algorithmic Reasoning with Language Models by Composing Decompositions." NeurIPS 2023. `(58 pages)`
>
> [13] "Uncovering the hidden dynamics of video self-supervised learning under distribution shifts." NeurIPS 2023. `(69 pages)`
>
> [14] "Principle-driven self-alignment of language models from scratch with minimal human supervision." NeurIPS 2023. `(55 pages)`
>
> [15] "WITRAN: Water-wave information transmission and recurrent acceleration network for long-range time series forecasting." NeurIPS 2023. `(68 pages)`
>
> ---
>
> Kindly note that the last 6 pages of our PDF are *NeurIPS Paper Checklist*, which did not exist last year. Therefore, they should not be counted towards the effective total page count.

---

### Official Review · Reviewer_x8xk · 2024-07-10

**Soundness:** 2
**Presentation:** 2
**Contribution:** 4
**Rating:** 8
**Confidence:** 3

**Summary:**

The paper introduces a method to compute exact conformal prediction intervals that have statistical guarantees. The method improves from previous work in three ways:
* The conformal interval has exact guarantees instead of approximative
* The loss function family covered is larger than convex, same for the regularizer family
* It works for linearly constrained problem

The core idea is to build a path from  value $t\in [0,1]$ to targets $y$ and find how the parameters of the function $\mathbf{w}$ evolve as we vary $t$ (and therefore $y$). The weights $\mathbf{w}(\cdot)$ as a function of $t$ satisfy an ODE that can be solve with standard solvers when the functions in the loss function and constrains are piecewise continuously differentiable.

The authors then show the improvement in terms of coverage, dataset size and more importantly speed on several datasets.

**Strengths:**

* The examples are compelling are show very well the improvement
* The contribution is pretty significant as it widens the possible families of loss functions

**Weaknesses:**

* My main issue is that the paper is very hard to read. It is too notation heavy and dense. For example, why use $\hat \theta$ instead of $\theta$? The $\hat {}$ make them look like estimators when it is not the case here
* Classification is a big application of conformal prediction, it is unfortunate that we do not have an example .
* Figure 1 might be a bit misleading. It seems to imply that the method works for higher dimensions of $\mathcal{Y}$ but it does not seem the case

**Questions:**

* Does the method require more memory than the baseline methods?
* I am not convinced by the explanation given for classification(1067-1074). This might be valid for some ranking tasks but not when there is no order in the classes, in my opinion. Could you illustrate it, even with a toy example?

**Limitations:**

* You mention not implementing [13] because it uses underlying structure and therefore might be faster. It would still be interesting to see if the gain of speed from using [13] instead of your method is significant or not, in cases where it applies.
* (see comment on classification)

---

> ### Author Rebuttal · Authors · 2024-08-07
>
> # Many thanks for acknowledging our research!
> Dear reviewer x8xk,
>
> We want to express our heartfelt thanks for taking the time to review our paper and for your kind words of appreciation and support for our work! In the following, we will provide a comprehensive response to your review comments.
>
> ---
> ## Ⅰ. Notation
> > It is too notation heavy and dense
>
> Thank you for your valuable comments. We understand and appreciate your concerns regarding the mathematical notation. **For a comprehensive explanation, please refer to our general response.**
>
> While the statistical derivations in the paper indeed necessitate a certain level of mathematical proficiency, we believe they will appeal to researchers from diverse fields. This, in turn, can foster collaboration between the communities of *conformal prediction* and *numerical optimization*.
> > why use $\hat{\theta}$ instead of $\theta$? The $\hat{~}$ make them look like estimators
>
> Thank you for your thought-provoking question! In our intention, $\hat{\theta}^k$ in the analysis is indeed used to denote **the estimator of ${\theta}^k$**. Technically,
>
> - Let us recall that Lemma 3 (page 20) explains that for the special case of $PC^r$ functions, there is a simpler expression for the Clarke subdifferential $\partial f$ (Definition 5) in terms of the gradients of the selection functions $\\{D^k\_f\\}$. Hence, the Clarke subdifferential of $f$ is easy to compute and that the set of nonsmooth points of $f$ can essentially be described as a level set of certain smooth functions (see our line 73), i.e., $\sum\_k \theta^k D^k\_f$, where the $f$ could be any function in Assumption 1 or 2.
>
> - Since the combination of $\theta^k$ can be non-unique (as long as they form a valid level set), it is challenging to use this set of non-vanishing coefficients to obtain the first-order approximations for nonsmooth analysis. However, in theory, we can compute a set of coefficients $\\{\hat{\theta}^k\\}$ (based on the analytical properties of $f$) to uniquely represent the convex hull of the gradients of the selection functions, i.e., we can have $\partial f=\sum\_k \hat{\theta}^k \nabla D^k\_f$.
>
> - A similar concept is: the model parameter $\mathbf{w}$ in Eq. (1) could be somehow arbitrary as long as it satisfies the constraint conditions, whereas $ \mathbf{w}^*$ is the estimator of $\mathbf{w}$, found as the optimal solution of the objective after training / fitting.
>
> For a more detailed introduction into nonsmooth analysis, we refer to [1]. Inspired by your question, **we are considering using Carathéodory's theorem to reduce the number of selection functions needed for computing the Clarke subdifferential**. Alternatively, **we may change it to a more concise notation, if it does not cause ambiguity in the main body.**
>
> ## Ⅱ. Label Space
> > Classification: it is unfortunate that we do not have an example
>
> - We do agree with your view that classification problems are an important application. In our current version, both the inverse Gaussian deviance and group lasso estimators **can be used for classification tasks as long as the labels in the dataset are categorical** (please refer to their respective original papers).
>
> - We primarily showcase regression tasks here because, in conformalization for regression tasks, the standard full CP computation is more demanding. **In regression tasks, the label space is dense rather than discrete, making it more representative and challenging** compared to classification tasks.
>
> > Figure 1 might be a bit misleading. It seems to imply that the method works for higher dimensions ...
>
> We sincerely appreciate this feedback, and we have added the $\mathcal{Y} \subseteq \mathbb{R}^1$ in its caption. We believe that readers will easily recognize this when reading Section 2 as well.
>
> Motivated by your comment, if $\mathcal{Y}$ is higher-dimensional (e.g., in multi-task learning), we cannot use a single scalar variable $z$ to loop over the entire space. In this case, our $y_{n+1}$ would be indexed by $\[z_1, \ldots, z_p\]$. The intuitive idea is that the homotopy path would turn into a **solution surface** [2], and our ODE system could be rewritten as a similar partial differential equation system. *While it introduces new challenges, it's very promising. Now we'll not expanding into this scenario, as it would significantly increase the notational complexity.*
>
> > Classification: ... when there is no order in the classes. Could you illustrate it?
>
> We are not senior researchers in classification theory, but if unordered classes cannot be encoded into a 1-dimensional $y$, then you're correct. Thanks again for your insights! It's very helpful in improving the quality of work.
>
> ## Ⅲ. Computations
> > Does the method require more memory than baseline?
>
> Yes, due to character limitation, we invite you to see Appendix F.3 for discussion. Meanwhile we believe that in most cases, the practical bottleneck of the conformalized algorithms is runtime cost rather than system memory.
> > the gain of speed from using [13] instead of your method
>
> Sorry for any confusion caused. The work [13] and our Algorithm 1 are completely identical in actual implementation, so there will be no gain in speed (if we dismiss the randomness). **Please refer to section ③ in our general rebuttal**.
> > You mention not implementing [13] because it uses underlying structure and therefore might be faster
>
> This was our oversight, and **we have now corrected the relevant descriptions** in Appendix D.3 (line 964). We apologize again for any confusion caused! We will continue to check for such details.
>
> ---
> **References:**
>
> [1] "Introduction to Nonsmooth Optimization: theory, practice and software." Springer, 2014
>
> [2] "Grouping pursuit through a regularization solution surface." JASA, 2010
>
> **We thank the Reviewer x8xk again for your insightful feedback and strong support of our submission! If you have any remaining concerns or further inquiries, please do not hesitate to let us know.**

---

> > ### Comment · Reviewer_x8xk · 2024-08-12
> >
> > Thanks for this extensive comment.
> >
> > First, I think there is some modification needed to be made following this discussion:
> > 1- removing the comments on classification / heavily modifying as your framework does not fit it yet
> > 2- Editing the figure 1 plot, because even if your mention $\mathbb{R}^1$, a reader skimming through the paper might not understand that it only works on one dimensional target
> > 3- It might be more interesting for the reader to know about the memory/runtime cost  challenges within the main core of the paper than buried deep in the appendix.
> >
> >
> > Moreover, I have to partly agree with **KUWB**: This paper is too lengthy making it hard to parse for both reviewers and future readers. You gave the example of simplified notations in appendix F.3 for the simpler case of classic losses. In my opinion, this should have been the one in the main paper, as this will be what interests 95% of readers. The results with the Clarke differential should have been in the appendix, making the main part of the paper easier to read and parse. I understand that the results using Clarke differential are more challenging, and you want to display it front page, but by providing the simpler case first, the reader can build a better intuition for what is happening underneath.
> >
> >
> > Finally, on top of your paper, you can also make the rebuttal process easier on readers/reviewers. Your comments are quite lengthy and could really be summarized. We mostly have multiple papers to review and comments on so it helps us when you stay brief and clear.
> >
> > I still believe the paper is worth publishing in NeurIPS, so I will keep my grade to 8 but it is a borderline 7 for all the reasons mentioned above.

---

> > > ### Author Response · Authors · 2024-08-13
> > >
> > > Dear reviewer x8xk,
> > >
> > > 1. We greatly appreciate your insights regarding the classification setting, and we will add a more restrictive description of the labels. We also commit to modifying Figure 1 based on your suggestion. Since all previous baselines were also unable to extend to more complex classification tasks (i.e. beyond $\mathbb{R}^1$), we will briefly outline our thoughts on how to achieve this extension in the future.
> > >
> > > 2. We will emphasize in the section of complexity analysis of the main body that our algorithm requires more memory.
> > >
> > > 3. Based on the feedback from several reviewers, in the next version, we will move minor technical parts from the main body to the appendix and bring some intuitive explanations from the appendix into the main body, which helps improve the overall readability of the paper. We have also reduced the content in the appendix to better highlight the main focus (see other rebuttals).
> > >
> > > We sincerely thank you once again for your generous support during the discussion phase!

---

### Official Review · Reviewer_4Aeq · 2024-07-10

**Soundness:** 4
**Presentation:** 2
**Contribution:** 4
**Rating:** 8
**Confidence:** 4

**Summary:**

In the manuscript "Towards fast exact Confomralization of Generalised Parametric Estimation" the authors provide a very interesting generalisation of an approach, fundamental even if a bit disregarded in the mainstream literature on Conformal Prediction, that aims at computing the whole solution path of a regression algorithm in order to render more efficient the estimation of (full) conformal prediction sets.
After having described the very general framework in which they operate, the authors proceed to obtain theoretical insights about their methodology. More specifically, they characterise the underlying structure of the path to be an inherently piecewise smooth function.
After doing so they propose a practical methodology to compute such path, essentially by solving an ODE.
The method is then evaluated with an extensive empirical study, showing its remarkable performance

**Strengths:**

- The work is extremely thorough and deep, and the discussion about the different peculiarities of their approach
- The referencing work is quite remarkable, as it depicts very clearly the literature landscape where the authors are moving
- While building on previous works, the authors provide a very novel take on the subject, proposing a completely new methodology, of great practical usability

**Weaknesses:**

- The authors are not tackling an easy task in terms of communication, having to tackle, and in a fairly deep way, many different topics. For this reason I find the presentation a bit "nebulous" and vague at times
- No code is provided, which has prevented me from verifying some of the claims.

**Questions:**

- Some claims are a bit vague. I understand why the authors focus on Full conformal, but I doubt a reader not fully aware of Conformal Prediction will be able to do it (Paragraph 2 of the introduction). I suggest to the authors to be more explicit about the reasons
- Footnote of page 1 - the authors need to clarify that symmetricity in this case is intended in the specific sense used in the CP literature, which in fact is defined later on... or omit the consideration which seems rather tangential
- Rather then "inexact" I would rather talk about lack of (finite sample) calibration.
- I don't understand why the already present approaches are deemed to be "black boxes". moreover I don't think that "opening the black box" is the right narrative to use in this case. Maybe something along the line of "let's generalise..."
- The label space is a generic $\mathcal{Y}$. I am wondering if the authors can be more specific about, for instance, possible extensions of their approach to the relatively rich field of Conformal Prediction for multivariate and complex data.

**Limitations:**

The authors state relevant open problems on the subject.
Since the very clear applicative interest this paper may have, I believe that adding something on the methodological side (such as the need for extensions to the multivariate case, or some ideas about possible extensions to a dependent data setting) could be very interesting.

---

> ### Author Rebuttal · Authors · 2024-08-07
>
> # Many thanks for acknowledging our work!
> Dear reviewer 4Aeq,
>
> We extend our sincere gratitude for dedicating your valuable time to reviewing our paper and for expressing your appreciation and support for our work! In the following, we will provide a comprehensive response to your questions.
>
> ---
>
> ## Ⅰ. Introduction
>
> >  I understand why the authors focus on Full conformal, but I doubt a reader not fully aware of Conformal Prediction will be able to do it (Paragraph 2 of the introduction).
>
> Your feedback is extremely beneficial to us.
>
> I believe **Reviewer jxtf** shares similar concerns. Please allow us to briefly describe the differences between the two approaches.
> - In SCP, we first split the entire dataset into a training set and a calibration set. On the training set, we use algorithm $\mathcal{A}$ to obtain the model parameter $\mathbf{w}^*$. Then, on the calibration set, we get prediction values using $\mathbf{w}^*$, and subsequently obtain conformity scores to calculate the conformal set. This process involves only one single fitting.
> - Full CP, on the other hand, avoids data splitting and does not discard any sample points. It loops over all possible $y_{n+1}$ values and for each new candidate, we refit the augmented data to compute the $\mathbf{w}^*$. Based on these solutions we obtain a set of training scores. Therefore, the traditional computation of full CP is significantly more expensive.
>
> > I suggest to the authors to be more explicit about the reasons
>
> Thank you for your constructive suggestion. I agree that we may omitted some relevant background in the introduction. We have added the necessary descriptions in the second paragraph of the "*1. Introduction*" on the first page to help readers better understand our considerations.
> > Footnote of page 1 - the authors need to clarify that symmetricity in this case is intended in the specific sense used in the CP literature ... or omit the consideration which seems rather tangential
>
> Thank you for your thoughtful feedback! We find your advice very reasonable, and after our careful consideration, we have decided to remove this footnote to reduce the technical density of the main body. We also agree that it does not directly relate to the main theme of this work.
>
> ## Ⅱ. Technical Phrase
>
> > Rather then "inexact" I would rather talk about lack of (finite sample) calibration
>
> Thanks for your insightful advice. We have revised the relevant statements on the second page.
>
> > I don't understand why the already present approaches are deemed to be "black boxes"
>
> This is our oversight; please refer to the general response, and sorry for the confusions.
>
> > I don't think that "opening the black box" is the right narrative to use in this case
>
> We have revised the relevant phrases according to the general rebuttal above. They now should be reasonable in the new context.
>
> If you have any further thoughts, please feel free to share them.
>
> ## Ⅲ. Potential Extension
>
> > The label space is a generic $\mathcal{Y}$ ... if the authors can be more specific
>
> Yes, in the current version of our analysis, an implicit setting is $\mathcal{Y} \subseteq \mathbb{R}$. In this work, $y_i\in\mathcal{Y}$ need to be unidimensional. However, we are happy to discuss various extensions; please see the following question.
>
> > for instance, possible extensions of their approach to the relatively rich field of Conformal Prediction for multivariate and complex data.
>
> We believe our work has the potential to extend CP to multivariate and complex data that you mentioned. Specifically,
>
> - If the data features include multiple interrelated variables, we can use well-studied techniques such as ODE discovery [1] to encode the correlations and synergistic effects between variables. If the labels are multidimensional (e.g., in multi-task learning), the label space $\mathcal{Y}$ will also be indexed by multiple independent parameters $(z_1, ..., z_p)$. In this case, the homotopy solution path would turn into a solution surface, and our ODE system can be rewritten as a similar partial differential equation system.
>
> - In complex environments, such as dynamic data or online conformal inference, we can leverage knowledge from our by-product as a starting point to adapt our framework to an online mode for parameter updates.
>
> For other types of data like network dataset, we can use neural ODEs [3] for model reparameterization, considering the network structure and connection patterns. We also believe this would be very interesting, as previous baseline work [2] has even begun to explore this direction (albeit in a somewhat different setting). More potential extensions will be left for future work. We believe this paper can serve as an important cornerstone, attracting subsequent researchers to fill these gaps.
>
> > the very clear applicative interest this paper may have
>
> We are very pleased to hear that you find our new algorithm to be widely applicable.
>
> > adding something on the methodological side (such as the need for extensions to the multivariate case, or some ideas about possible extensions to a dependent data setting) could be very interesting.
>
> Thank you for the helpful comments. We agree with your viewpoint, and inspired by your suggestion, we will add discussions on extensions in our Section 6 or Appendix F. We also hope to attract researchers from other relevant communities.
>
> ---
> **References:**
>
> [1] "Discovering governing equations from data by sparse identification of nonlinear dynamical systems." PNAS, 2016
>
> [2] "Fast exact conformalization of the lasso using piecewise linear homotopy." Biometrika, 2019
>
> [3] "Neural Ordinary Differential Equations." NeurIPS 2018
>
> **We thank the Reviewer 4Aeq again for your insightful feedback to improve our paper quality and strong support! If you have any remaining concerns or further inquiries, please do not hesitate to let us know.**

---

> > ### Comment · Reviewer_4Aeq · 2024-08-07
> >
> > I am fully satisfied with the provided replies, and I believe the contribution to be worthy of acceptance.

---

> > > ### Author Response · Authors · 2024-08-07
> > > **response**
> > >
> > > Dear reviewer 4Aeq,
> > >
> > > We are very pleased that our response has addressed your questions. Thank you again for appreciating our technical contributions!

---

### Official Review · Reviewer_ji5B · 2024-07-12

**Soundness:** 3
**Presentation:** 2
**Contribution:** 3
**Rating:** 6
**Confidence:** 2

**Summary:**

The authors introduce an algorithm to compute prediction intervals for conformal prediction in the context of empirical risk minimization  (ERM) with constraints on the parameters. To do so, the authors derive a differential equation solved by the ERM estimator as a function of the last label to compute a solution path and generate the conformity scores for each possible label.
In their numerical experiments, the method is compared to the baseline (which simply iterates over a grid of labels) and split conformal prediction (SCP). The authors observe that their method provide the correct coverage and roughly the same interval length as the baseline for a fraction of the numerical cost.

**Strengths:**

- The method introduced by the authors works for general scenarios, extending beyond the cases of Ridge / Lasso.
- On the numerical side, their methods appear to provide a significant improvement (in terms of running time) over the baseline while having the same interval size. As a by product, the authors show that their method can be used to update the estimator when a single label is changed in the training data.

**Weaknesses:**

- Clarity : I found the mathematical details of section 3 to be difficult to follow and check (for the parts in the Appendix). I feel like some parts, while they are important for the rigor of the paper, could be explained after the introduction of Algorithm 1 or put in the Appendix. For instance Theorem 4 and Theorem 6.

**Questions:**

- It would be nice to have a comparison with the running time of split conformal prediction (while the computations are not exactly the same) as in Figure 3 to get a sense of the order of magnitude of the computation time.
- For simple settings such as Lasso, it would have been nice to compare the computation time with methods from the literature e.g. [Lei, 2017].
- How does the method scale (in terms of running time) with the dimensionality ?

Misc. questions :
- In equation (12), is there a difference between Q(w*) and Q(w* | z) from (9) or is the conditioning on z implicit ?

**Limitations:**

Some limitations are discussed in Appendix F. As the paper is quite theoretical, the question of societal impact is not applicable here.

---

> ### Author Rebuttal · Authors · 2024-08-07
>
> # Many thanks for appreciating our research!
> Dear reviewer ji5B,
>
> We are truly grateful for your thoughtful review of our paper and for the encouraging support you've shown towards our work! In the following, we will provide a comprehensive response to your comments.
>
> ---
> > can be used to update the estimator when a single label is changed in the training data
>
> Thank you for your interest in our by-product. In fact, it allows for the simultaneous variation of multiple labels. These indices can form a subset $\mathcal{D}$ (see line 320), making our Algorithm 2 more efficient.
>
> >  I feel like some parts ... could be explained after the introduction of Algorithm 1 or put in the Appendix
>
> - Thanks for your comments, and we fully understand your concerns.
>
>   - Regarding the burden of mathematical notation, we invite you to refer to our general response. As for the statistical derivations in the appendix, we acknowledge that they require a certain level of mathematical maturity to fully understand. However, we believe this will **attract interest from researchers in other fields** and help **bridge the communities of conformal prediction and numerical optimization**.
>
>   - Theorems 4 and 6 are crucial supports for our algorithm. Theorem 4 provides the foundational guarantee for Theorem 5, ensuring the computational feasibility of our core idea, which is essential for designing Algorithm 1. Theorem 6 is closely related to Corollary 1, offering geometric insights into the path and forming the unified view that we have claimed in the abstract. We consider these theorems indispensable parts of our contributions.
>
> - By your comments, we **deleted some minor technical paragraphs, and enhanced our descriptions to better link the theorems and algorithms, ensuring readers do not get lost.** We appreciate your suggestion!
>
> > a comparison with the running time of SCP as in Figure 3 to get a sense of the order of magnitude
>
> Thank you for your constructive feedback! Our thoughts are as follows:
>
> - This work primarily focuses on optimization for full CP rather than SCP, so comparing efficiency (runtime) with SCP is not particularly meaningful.
>
> - SCP is theoretically guaranteed to be faster than standard full CP and our algorithm because **it only requires a single training on the training set** and then calculates conformity scores on calibration set [1]. In contrast, our algorithm needs to compute the solution spectrum w.r.t. entire label space, involving repeated calls to and fitting on the training set. The **order of magnitude can be easily estimated**; previous observations & analyses (e.g. [2]) have shown it to be approximately $\frac{1}{N_{\text{gird}}}$ times that of standard full CP, which already provided in our results.
>
> - Figures 3/4 are already content-rich; adding more curves might reduce the readability of the results.
>
> > compare the computation time with methods from [Lei, 2017].
>
> Please refer to general rebuttal, section ③. Thank you!
>
> > How does the method scale (in terms of running time) with the dimensionality?
>
> - Thank you for your insightful question. In Algorithm 1, we essentially performed two things: one is numerical integration over $[z_{min}, z_{max}]$, and the other is refreshing the current partitions $\mathcal{N}\_\text{E}$, $\mathcal{P}\_\text{E}$, $\mathcal{P}\_\text{I}$, $I_L$, $I_\Omega$ at the kinks.
>
>   - Let the dimensionality be $p$, the complexity of the latter is $\mathcal{O}(p)$, as we analyzed it in Section 4.3.
>
>   - The complexity of the former when varying $p$, however, is somewhat nontrivial. This is primarily due to a fact that the computational complexity of numerical solvers for ODEs *can indeed differ significantly* based on the employed method and the particularities of the equations at hand. For example, factors like the system's stiffness and the length (or area) of the solving interval, can play a crucial role [3]. Without further assumptions, it is difficult to analyze the complexity of numerically solving the entire ODE system. However, recent studies (e.g., [4]) have increasingly shown that oracle complexity (a.k.a. query complexity) plays a very significant role throughout the computation. Using our method under the sweep operator and Sherman–Morrison formula (see Section 4.1, Appendix B.8), the upper bound on oracle complexity is at most $\mathcal{O}(p^2)$.
>
> - Due to the adaptiveness of numerical solvers, it is challenging to establish an exact / tight relationship between overall runtime and $p$. We believe **it should lie between linear and quadratic growth**.
>
> - Given the suggestions of other reviewers, we temporarily don't intend to further expand the appendix length (e.g., adding more experiments).
>
> > the difference between Q(w*) in (12) and Q(w* | z) from (9) or is the conditioning on z implicit ?
>
> Your understanding is correct; they all refer to the same block matrix. When presenting our Theorem 5, we use $\mathbf{Q}(\mathbf{w}^\star)$ to represent $\mathbf{Q}(\mathbf{w}^\star|z)$, and $\mathbf{D}^{\prime}(\mathbf{w}^\star)$ to represent $\mathbf{D}^{\prime}(\mathbf{w}^\star|z)$. **This is to reduce the symbol density in the main body** without causing ambiguity.
>
> Note in appendix, we still use the full notation of them (see line 773) due to the needs of the analysis.
>
> ---
> **References:**
>
> [1]  "Conformal prediction: A gentle introduction." Foundations and Trends in Machine Learning, 2023
>
> [2] "Fast exact conformalization of the lasso using piecewise linear homotopy." Biometrika, 2019
>
> [3] "Diagonally implicit Runge–Kutta methods for stiff ODEs." Applied Numerical Mathematics, 2019
>
> [4] "A characterization of functions over the integers computable in polynomial time using discrete ordinary differential equations." Computational Complexity, 2023
>
> **We thank the Reviewer ji5B again for your insightful feedback and strong support of our submission! If you have any remaining concerns or further inquiries, please do not hesitate to let us know.**

---

### Author Rebuttal · Authors · 2024-08-07

# Gratitude to All the Reviewers 😃
---

Dear Reviewers,

Thanks for the time and effort you have devoted to evaluating our submission #451. We also wish to express our appreciation for your recognition of the strengths of this work, including:

> (ji5B) The method introduced by the authors **works for general scenarios**, extending beyond the cases of Ridge / Lasso

> (ji5B)  their methods appear to **provide a significant improvement** (in terms of running time) over the baseline while having the same interval size

> (4Aeq) the authors provide **a very interesting generalisation** of an approach, fundamental even if a bit disregarded in the mainstream literature on CP

> (4Aeq) The work is **extremely thorough and deep**, and the discussion about the different peculiarities of their approach

> (4Aeq)The **referencing work is quite remarkable**, as it depicts very clearly the literature landscape where the authors are moving

> (4Aeq) the authors provide a very novel take on the subject, proposing a completely new methodology, of **great practical usability**

> (x8xk) The **contribution is pretty significant** as it widens the possible families of loss functions

> (x8xk) The conformal interval **has exact guarantees** instead of approximative; The loss function family **covered is larger than convex**, same for the regularizer family

> (x8xk) The **examples are compelling** ... show very well the improvement

> (KUWB) This paper **studies an important problem** in conformal inference

> (KUWB) The proposed method could potentially **reduce the computational burden**

> (jxtf) Characterizing the solution path of ... optimization problems **is relevant, even beyond the CP framework**

> (jxtf) The Lagrangian reparametrization approach **is interesting**

> (jxtf) The proposed method seems to be more **efficient than** Split CP

You have also raised some questions/concerns, to which we have replied in our detailed rebuttal. Here we provide responses to several recurring concerns as follows

### ① Notation is dense / heavy

$\quad$*(Reviewer ji5B, Reviewer x8xk, Reviewer KUWB)*

We extend our gratitude to the reviewers for this valuable feedback. This work is not solely a contribution at the algorithmic level; it also provides new statistical insights and an analytical framework in the optimization theory. NeurIPS hosts a diverse community, drawing participants from a broad range of backgrounds; we believe our work could attract researchers not only from the field of conformal prediction but also from *optimization* and *learning theory*.

And unfortunately, **under the premise of ensuring mathematical rigor and technical correctness, it appears that further simplification of symbols / theorems is not feasible currently**. It is worth noting that our initial draft was significantly more complex than the current one, and the present version is likely the most concise iteration we have achieved through repeated exploration. To achieve this goal, our efforts have included:
1) When deriving the ODE system (11) or (14), we initially arrived at complex preliminary conclusions, with formulas at least twice the length of the current ones. We simplified the obtained expressions by utilizing matrix computation properties and non-convex analysis, enhancing their human readability. This is reflected in **page 24 to 25** of our proof.
2) We have carefully considered how to present our notation system and the main core theorem outcomes. We defined **a certain matrix inversion** (9), which includes many of the common structures used in theorems, significantly reducing the notation density in the paper.
3) To enhance readers' understanding of the framework presented, we provide a more in-depth discussion and explore the potential for simplifying notation under stronger assumptions. You are welcome to visit our **Appendix F.2** for discussions on this point. We also applied it to two simple toy cases (*i.e.*, vanilla Lasso and elastic net) in Appendix C.

As you suggested, conciseness and readability remain our goals, and we are continuously contemplating how to further improve in this regard.

### ② On the “open the black-box”

$\quad$*(Reviewer 4Aeq, Reviewer jxtf)*

We appreciate the reviewers' careful reading and for pointing this out. We realize that our current phrasing is incorrect.

Our use of "black-box approach" should indeed refer to grid-search based methods (*i.e.*, existing batch trainings on different $y_{n+1}$), as they completely disregard the underlying potential path structure. "Black-box approach" should not describe the fast conformalization methods (*i.e.* baseline in Table 1; whether precise or imprecise), as they all reformulate the conformalization problem as a path-following optimization  problem and use white-box methods to compute solution spectrum. **We have corrected the relevant phrasing.** Thanks!

### ③ Experiment: conformalization of Lasso

$\quad$*(Reviewer ji5B, Reviewer x8xk)*

Several reviewers suggested we compare our work with the Lasso conformalization algorithm from (Lei, 2019) [13]. We appreciate this feedback, but we would like to politely point out that Lasso inherently falls under generalized parametric estimation (see Table 1). Thus, (Lei, 2019) [13] is **a special case within our framework**. Further technical analysis is available in Appendix C.1, which demonstrates how our framework recovers the piecewise linear homotopy path when specifying the loss to be quadratic and the regularizer to be $\ell_1$-norm.

**From a practical perspective, these two algorithms are essentially the same**, making the comparison potentially meaningless. We apologize for not clarifying this in our experimental setup and have revised the confusing descriptions in the appendix. Thanks!

---

Please feel free to let us know if you still have any remaining concern, we will be happy to address them. Thanks for your time and support!

$ $

Best Regards,

Submission451 Authors

---

### Decision · Program_Chairs · 2024-09-25

**Decision:**

Accept (poster)

**Comment:**

There is an overall agreement among the evaluators about the value of the result presented in this paper. The reviews and the discussion were detailed and provided many directions to improve the presentation further. The paper provides a valuable contribution towards efficient conformal prediction, a promising approach to uncertainty estimation in many practical settings. I recommend acceptance as a poster.